# Scheduled Intrinsic Drive: A Hierarchical Take on Intrinsically Motivated Exploration

## Abstract

Exploration in sparse reward reinforcement learning remains an open challenge. Many state-of-the-art methods use intrinsic motivation to complement the sparse extrinsic reward signal, giving the agent more opportunities to receive feedback during exploration. Commonly these signals are added as bonus rewards, which results in a mixture policy that neither conducts exploration nor task fulfillment resolutely. In this paper, we instead learn separate intrinsic and extrinsic task policies and schedule between these different drives to accelerate exploration and stabilize learning. Moreover, we introduce a new type of intrinsic reward denoted as *successor feature control* (**SFC**), which is general and not task-specific. It takes into account statistics over complete trajectories and thus differs from previous methods that only use local information to evaluate intrinsic motivation. We evaluate our proposed *scheduled intrinsic drive* (**SID**) agent using three different environments with pure visual inputs: VizDoom, DeepMind Lab and DeepMind Control Suite. The results show a substantially improved exploration efficiency with SFC and the hierarchical usage of the intrinsic drives. A video of our experimental results can be found at `https://gofile.io/?c=HpEwTd`.

## 1 Introduction

Reinforcement learning (RL) agents learn on evaluative feedback (reward signals) instead of instructive feedback (ground truth labels), which takes the process of automating the development of intelligent problem-solving agents one step further (Sutton & Barto, 2018). With deep networks as powerful function approximators bringing traditional RL into high-dimensional domains, deep reinforcement learning (DRL) has shown great potential (Mnih et al., 2015; 2016; Schulman et al., 2017; Horgan et al., 2018). However, the success of DRL often relies on carefully shaped dense extrinsic reward signals. Although shaping extrinsic rewards can greatly support the agent in finding solutions and shortening the interaction time, designing such dense extrinsic signals often requires substantial domain knowledge, and calculating them typically requires ground truth state information, both of which is hard to obtain in the context of robots acting in the real world. When not carefully designed, the reward shape could sometimes serve as bias or even distractions and could potentially hinder the discovery of optimal solutions. More importantly, learning on dense extrinsic rewards goes backwards on the progress of reducing supervision and could prevent the agent from taking full advantage of the RL framework.

In this paper, we consider terminal reward RL settings, where a signal is only given when the final goal is achieved. When learning with only an extrinsic terminal reward indicating the task at hand, intelligent agents are given the opportunity to potentially discover optimal solutions even out of the scope of the well established domain knowledge.

However, in many real-world problems defining a task only by a terminal reward means that the learning signal can be extremely sparse. The RL agent would have no clue about what task to accomplish until it receives the terminal reward for the first time by chance. Therefore in those scenarios guided and structured exploration is crucial, which is where intrinsically-motivated exploration (Oudeyer & Kaplan, 2008; Schmidhuber, 2010) has recently gained great success (Pathak et al., 2017; Burda et al., 2018b). Most commonly in current state-of-the-art approaches, an intrinsic reward is added as a reward bonus to the extrinsic reward. Maximizing this combined reward signal, however, results in a mixture policy that neither acts greedily with regard to extrinsic reward max-

imization nor to exploration. Furthermore, the non-stationary nature of the intrinsic signals could potentially lead to unstable learning on the combined reward. In addition, current state-of-the-art methods have been mostly looking at local information calculated out of 1-step lookahead for the estimation of the intrinsic rewards, e.g. one step prediction error (Pathak et al., 2017), or network distillation error of the next state (Burda et al., 2018b). Although those intrinsic signals can be propagated back to earlier states with temporal difference (TD) learning, it is not clear that this results in optimal long-term exploration. We seek to address the aforementioned issues as follows:

1. We propose a hierarchical agent *scheduled intrinsic drive* (**SID**) that focuses on one motivation at a time: It learns two separate policies which maximize the extrinsic and intrinsic rewards respectively. A high-level scheduler periodically selects to follow either the extrinsic or the intrinsic policy to gather experiences. Disentangling the two policies allows the agent to faithfully conduct either pure exploration or pure extrinsic task fulfillment. Moreover, scheduling (even within an episode) implicitly increases the behavior policy space exponentially, which drastically differs from previous methods where the behavior policy could only change slowly due to the incremental nature of TD learning.

2. We introduce *successor feature control* (**SFC**), a novel intrinsic reward that is based on the concept of successor features. This feature representation characterizes states through the features of all its successor states instead of looking at local information only. This implicitly makes our method temporally extended, which enables more structured and farsighted exploration that is crucial in exploration-challenging environments.

We note that both the proposed intrinsic reward SFC and the hierarchical exploration framework SID are without any task-specific components, and can be incorporated into existing DRL methods with minimal computation overhead. We present experimental results in three sets of environments, evaluating our proposed agent in the domains of visual navigation and control from pixels, as well as its capabilities of finding optimal solutions under distraction.

## 2 RELATED WORK

**Intrinsic Motivation and Auxiliary Tasks**     Intrinsic motivation can be defined as agents conducting actions purely out of the satisfaction of its internal rewarding system rather than the extrinsic rewards (Oudeyer & Kaplan, 2008; Schmidhuber, 2010). There exist various forms of intrinsic motivation and they have achieved substantial improvement in guiding exploration for DRL, in tasks where extrinsic signals are sparse or missing altogether.

(Pathak et al., 2017) proposed to evaluate curiosity, one of the most widely used kinds of intrinsic motivation, with the 1-step prediction error of the features of the next state made by a forward dynamics model. Their ICM module has been shown to work well in visual domains including first-person view navigation. Since ICM is potentially susceptible to stochastic transitions (Burda et al., 2018a), Burda et al. (2018b) propose as a reward bonus the error of predicting the features of the current state output by a randomly initialized fixed embedding network. The value function is decomposed for extrinsic and intrinsic reward, but different to us a single mixture policy is learned. Another form of curiosity, learning progress or the change in the prediction error, has been connected to count-based exploration via a pseudo-count (Bellemare et al., 2016; Ostrovski et al., 2017) and has also been used as a reward bonus. Savinov et al. (2018) propose to train a reachability network, which gives out a reward based on whether the current state is reachable within a certain amount of steps from any state in the current episode. Similar to our proposed SFC, their intrinsic motivation is related to choosing states that could lead to novel trajectories. However, we use two different distance metrics, theirs is explicitly learned to be proportional to the time step differences while ours is based on successor features which measures two states by the difference of the average feature activations of future trajectories. Moreover, their method rewards states with high distance to the states in the current episode while our method rewards states with high distance to the states also from past trajectories, as the successor features are trained from samples of the replay buffer.

Auxiliary tasks have been proposed for learning more representative and distinguishable features. Mirowski et al. (2016) add depth prediction and loop closure prediction as auxiliary tasks for learning the features. Jaderberg et al. (2016) learn separate policies for maximizing pixel changes (pixel control) and activating units of a specific hidden layer (feature control). However, their proposed

UNREAL agent never follows those auxiliary policies as they are only used to learn more suitable features for the main extrinsic task.

**Hierarchical RL**  Various HRL approaches have been proposed (Kulkarni et al., 2016a; Bacon et al., 2017; Vezhnevets et al., 2017; Krishnan et al., 2017). In the context of intrinsic motivation, feature control (Jaderberg et al., 2016) has been adopted into a hierarchical setting (Dilokthanakul et al., 2017), in which options are constructed for altering given features. However, they report that a flat policy trained on the intrinsic bonus achieves similar performance to the hierarchical agent.

Our hierarchical design is perhaps inspired mostly by the work of Riedmiller et al. (2018). Unlike other HRL approaches that try to learn a set of options (Sutton et al., 1999) to construct the optimal policy, their proposed SAC agent aims to learn one flat policy that maximizes the extrinsic reward. While SAC schedules between following the extrinsic task and a set of pre-defined auxiliary tasks such as maximizing touch sensor readings or translation velocity, in this paper we investigate scheduling between the extrinsic task and intrinsic motivation that is general and not task-specific. A concurrent work along this line is presented by Beyer et al. (2019).

**Successor Representation**  The successor representation (SR) was first introduced to improve generalization in TD learning (Dayan, 1993). While previous works extended SR to the deep setting for better generalized navigation and control algorithms across similar environments and changing goals (Kulkarni et al., 2016b; Barreto et al., 2017; Zhang et al., 2017), we focus on its temporarily extended property to accelerate exploration.

SR has also been investigated under the options framework. Machado et al. (2017); Tomar* et al. (2019) evaluate successor features with random policies to discover bottlenecks or landmarks based on the clustering of such features. Options are then learned to navigate to those sub-goals. However, it remained unclear if the options framework would help in sparse exploration setups.

When using SR to measure the intrinsic motivation, the most relevant work to ours is that of Machado et al. (2018). They also design a task-independent intrinsic reward based on SR, however they rely on the concept of count-based exploration and propose a reward bonus, that vastly differs from ours. Their bonus is inverse proportional to the norm of the SR while our formulation rewards change in the SR of two successive states. We will present our proposed method in the next section.

## 3  METHODS

We use the RL framework for learning and decision-making under uncertainty. It is formalized by Markov decision processes (MDPs) defined by the tuple $\langle \mathcal{S}, \mathcal{A}, p, r, \gamma \rangle$. At time step $t$ the agent samples an action $a \in \mathcal{A}$ according to policy $\pi(\cdot|s)$, which depends on its current state $s \in \mathcal{S}$. The agent receives a scalar reward $r \in \mathbb{R}$ and transits to the next state $s' \in \mathcal{S}$. The distribution of the corresponding state, action and reward process $(S_t, A_t, R_{t+1})$ is determined by the distribution of the initial state $S_0$, the transition operator $p$ and the policy $\pi$. The goal of the agent is to find a policy that maximizes the expectation of the sum of discounted rewards $\sum_{k=0}^{T} \gamma^k R_{t+k+1}$. We seek to speed up learning in sparse reward RL, where the reward signal is uninformative for almost all transitions. We set the focus on terminal reward scenarios, where the agent only receives a single reward of $+1$ for successfully accomplishing the task and $0$ otherwise.

We will first introduce our proposed intrinsic reward *successor feature control* (**SFC**) (3.1,3.2), then present our proposed hierachical framework for accelerating intrinsically motivated exploration, which we denote as *scheduled intrinsic drive* (**SID**) (Sec.3.3,3.4).

### 3.1  SUCCESSOR DISTANCE METRIC

In order to encode long-term statistics into the design of intrinsic rewards for far-sighted exploration, we build on the formulation of *successor representation* (SR), which introduces a temporally extended view of the states. Dayan (1993) introduced the idea of representing a state $s$ by the occupancies of all other states from a process starting in $s$ following a fixed policy $\pi$, where the occupancies denote the average number of time steps the state process stays in each state per episode. *Successor*

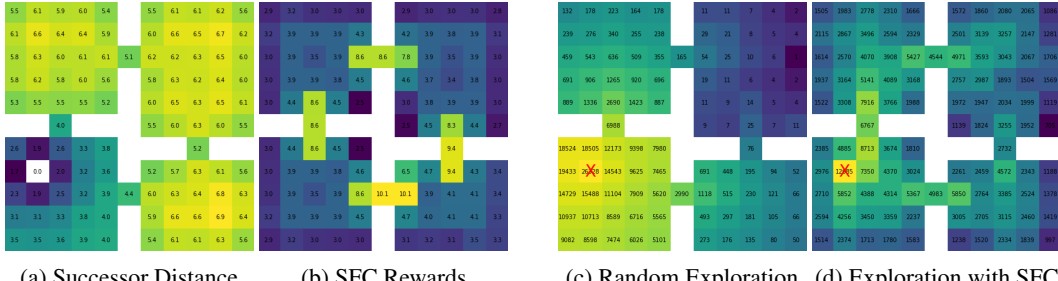

(a) Successor Distance     (b) SFC Rewards     (c) Random Exploration    (d) Exploration with SFC

Figure 1: The four-room domain (Sutton et al., 1999). The agent starts at the red cross and transitions to an adjacent state at each time step. The goal is to explore the four rooms when no extrinsic reward is provided. In **a)** each state is annotated by its SD (Eq.3) to the starting state and **b)** shows for each state the highest possible SFC reward (Eq.4) for a one-step transition from it. Here the successor features are learned using a random walk. **c)** and **d)** show a comparison between visitation counts of each state from a random agent and an agent that uses the SFC rewards for control via Q-learning. In the latter case the successor features are learned from scratch via TD.

In this environment, the agent receives high rewards for crossing bottleneck states, when the SF are learned beforehand, using a random policy. But even when the SF are learned during exploration, bottleneck states are still visited disproportionately high. Furthermore the intrinsic reward greatly improves exploration compared to a random agent. For implementation details see Appendix. D.4

*features* (SF) (Kulkarni et al., 2016b; Barreto et al., 2017) extend the concept to an arbitrary feature embedding $\phi : \mathcal{S} \to \mathbb{R}^m$. For a fixed policy $\pi$ and embedding $\phi$ the SF is defined by the $|m|$-dimensional vector

$$\psi_{\pi,\phi}(s) := \mathbb{E}_\pi \left[ \sum_{t=0}^{\infty} \gamma^t \phi(S_t) \Big| S_0 = s \right]. \tag{1}$$

Analogously, the SF represent the average discounted feature activations, when starting in $s$ and following $\pi$. They can be learned by *temporal difference* (TD) updates

$$\psi_{\pi,\phi}(S_t) \leftarrow \psi_{\pi,\phi}(S_t) + \alpha \Big[ \phi(S_t) + \gamma \psi_{\pi,\phi}(S_{t+1}) - \psi_{\pi,\phi}(S_t) \Big]. \tag{2}$$

SF have several interesting properties which make them appealing as a basis for an intrinsic reward signal: 1) They can be learned even in the absence of extrinsic rewards and without learning a transition model and therefore combine advantages of model-based and model-free RL (Stachenfeld et al., 2014). 2) They can be learned via computationally efficient TD. 3) They capture the expected feature activations for complete episodes. Therefore they contain information even of spatially and temporally distant states which might help for effective far-sighted exploration. Given the discussion, we introduce the successor distance (SD) metric that measures the distance between states by the similarity of their SF

$$d_{\pi,\phi}(s, s') := ||\psi_{\pi,\phi}(s) - \psi_{\pi,\phi}(s')||_2. \tag{3}$$

Fig.1 a) shows an example of the successor distance metric in the tabular case. There the SD roughly correlates to the length of the shortest path between the states. Using this metric to evaluate the intrinsic motivation, one choice could be to use the SD to a fixed anchor state as the intrinsic reward, which depends heavily on the anchor position. Even when a sensible choice for the anchor can be found, e.g. the initial state of an episode, the SDs of distant states from the anchor assimilate.

For a pair of states with a fixed spatial distance, their SD is higher when they are located in different rooms and the SD increases substantially when crossing rooms. Therefore the metric might capture the connectivity of the underlying state space.

## 3.2 SUCCESSOR FEATURE CONTROL

This observation motivates us to define the intrinsic reward *successor feature control* (SFC) as the squared SD of a pair of consecutive states

$$R_{t+1}^{\mathrm{sfc}} := ||\psi_{\pi,\phi}(S_{t+1}) - \psi_{\pi,\phi}(S_t)||_2^2. \tag{4}$$

A high SFC reward indicates a big change in the future feature activations when $\pi$ is followed. We argue this big change is a strong indicator of bottleneck states, since in bottlenecks a minor change in the action selection can lead to a vastly different trajectory being taken. Fig.1b) shows that those highly rewarding states under SFC and the true bottlenecks agree, which can be very valuable for exploration (Lehnert et al., 2018).

## 3.3 SCHEDULED INTRINSIC DRIVE

The classical way of adding the intrinsic reward to the extrinsic reward has several drawbacks. First, the final policy is not trained to maximize the actual objective but a mixed version. Second, the intrinsic reward signal is usually changing over time. Including this non-stationary signal in the overall reward can make learning of the actual task unstable. Furthermore, the performance is often extremely sensitive to the scaling of the intrinsic reward relative to the extrinsic and hence it has to be tuned very carefully for every environment.

To overcome these issues we propose scheduled intrinsic drive (SID), which learns two separate policies, one for each reward signal. During each episode the scheduler samples several times which of the two policies to follow for the next time steps. Each policy is trained off-policy from all of the transitions irrespective of which policy collected the data.

As SID does not add the two reward signals no scaling parameter is needed. A policy is learned that exclusively maximizes extrinsic reward and hence neither the final policy nor the learning process is disturbed by the intrinsic reward. At the same time exploration is ensured as there is experience collected by the policy that learns from the intrinsic reward. Furthermore, scheduling can help exploration as each policy is acted on for an extended time interval, allowing long-term exploration instead of local exploration. Besides that the agent is less susceptible to always go to a nearby small reward instead of looking for other larger rewards that maybe further away. A mixture policy might be attracted to the small reward while with SID the exploration policy is followed for several timesteps which can bring the agent to new states with larger rewards that it did not know of before.

We investigated several types of high-level schedulers, however, none of them consistently outperforms a random one. We present possible explanations why a random scheduler already performs well and present them in Appendix F along with the different scheduler choices we tested.

## 3.4 ALGORITHM IMPLEMENTATION

Our proposed method can be combined with an any approach that allows off-policy learning. This section describes an instantiation of the SID framework when using Ape-X DQN as a basic off-policy DRL algorithm Horgan et al. (2018) with SFC as the intrinsic reward, which we used for all experiments. We depict this algorithm instance in Appendix Figure 8 and more details are provided in Appendix C. The algorithm is composed of:

- A Q-Net $\{\theta_\varphi, \theta_E, \theta_I\}$: Contains a shared embedding $\theta_\varphi$ and two Q-value output heads $\theta_E$ (extrinsic) and $\theta_I$ (intrinsic).

- A SF-Net $\{\theta_\phi, \theta_\psi\}$: Contains an embedding $\theta_\phi$ and a successor feature head $\theta_\psi$. $\theta_\phi$ is initialized randomly and kept fixed during training. The output of SF-Net is used to calculate the SFC intrinsic reward (Eq.4). The SF-net is trained with the samples of the replay buffer, which contains the experience generated by the behavior policy.

- A high-level scheduler: Instantiated in each actor, selects which policy to follow (extrinsic or intrinsic) after a fixed number of environment steps (max episode length$/M$). The scheduler randomly picks one of the tasks with equal probability.

- $N$ parallel actors ($N = 8$): Each actor instantiates its own copy of the environment, periodically copies the latest model from the learner. We learn from $K$-step targets ($K = 5$), so each actor at each environment step stores $(s_{t-K}, a_{t-K}, \sum_{k=1}^{K} \gamma^{k-1} r_{t-K+k}, s_t)$ into a shared replay buffer. Each actor will act according to either the extrinsic or the intrinsic policy based on the current task selected by its scheduler.

- A learner: Learns the Q-Net ($\theta_E$ and $\theta_I$ are learned with the extrinsic and intrinsic reward respectively) and the SF-Net from samples (Eq.2) from the same shared replay buffer, which contains all experiences collected from following different policies.

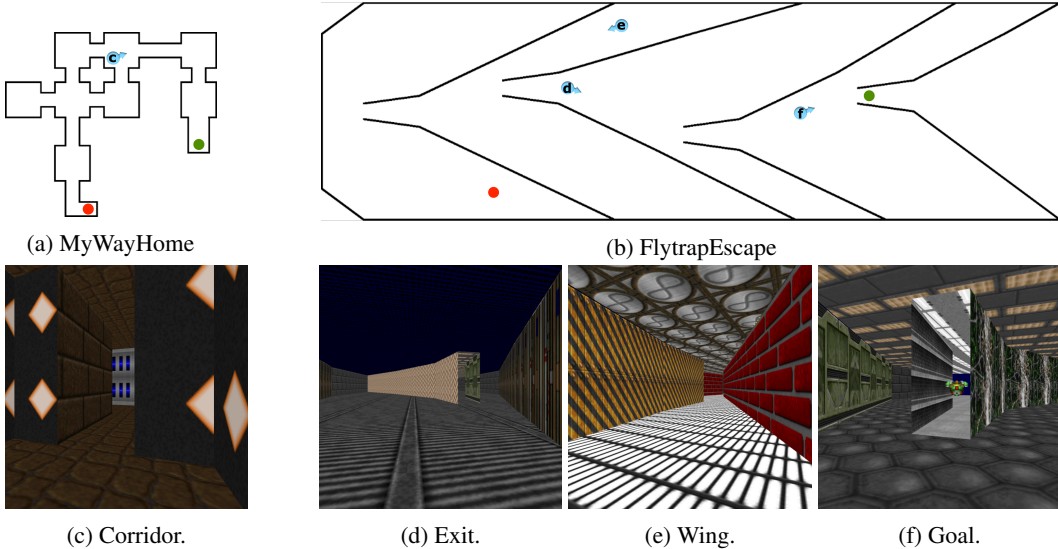

(a) MyWayHome              (b) FlytrapEscape

(c) Corridor.          (d) Exit.          (e) Wing.          (f) Goal.

Figure 2: VizDoom environments we evaluated on. 2a and 2b show the top-down views of My-WayHome and FlytrapEscape with the same downscaling ratio, with red dots marking the starting locations, green dots indicating the goal locations; 2c and 2d to 2f show exemplary first-person views captured from the marked poses (blue dots with arrows) from those two maps respectively.

## 4 EXPERIMENTS

We evaluate our proposed intrinsic reward SFC and the hierarchical framework of intrinsic motivation SID in three sets of simulated environments: VizDoom (Kempka et al., 2016), DeepMind Lab (Beattie et al., 2016) and DeepMind Control Suite (Tassa et al., 2018). Throughout all experiments, agents receive as input only raw pixels with no additional domain knowledge or task specific information. We mainly compare the following agent configurations: **M**: Ape-X DQN with 8 actors, train with only the extrinsic main task reward; **ICM**: train a single policy with the ICM reward bonus (Pathak et al., 2017); **RND**: train a single policy with the RND reward bonus (Burda et al., 2018b); **Ours**: with our proposed SID framework, schedule between following the extrinsic main task policy and the intrinsic policy trained with our proposed SFC reward.

We carried out an ablation study, where we compare the performance of an agent with intrinsic and extrinsic reward summed up, to the corresponding SID agent for each intrinsic reward type (ICM, RND, SFC). We present the plots and discussions in Section 4.4 Appendix A.

For the intrinsic reward normalization and the scaling for the extrinsic and intrinsic rewards we do a parameter sweep for each environment (Appendix C.4) and choose the best setting for each agent. We notice that our scheduling agent is much less sensitive to different scalings than agents with added reward bonus. Since our proposed SID setup requires an off-policy algorithm to learn from experiences generated by following different policies, we implement all the agents under the Ape-X DQN framework Horgan et al. (2018). After a parameter sweep we set the number of scheduled tasks per episode to $M = 8$ for our agent in all experiments, meaning each episode is divided into up to 8 sub-episodes, and for each of which either the extrinsic or the intrinsic policy is sampled as the behavior policy. Appendix C and D contain additional information about experimental setups and model training details.

### 4.1 VIZDOOM: SPARSE NAVIGATION

We start by verifying our implementation of the baseline algorithms in "DoomMyWayHome" which was previously used in several state-of-the-art intrinsic motivation papers (Pathak et al., 2017; Savinov et al., 2018). The agent needs to navigate based only on first-person view visual inputs through 8 rooms connected by corridors (Fig.2a), each with a distinct texture (Fig.2c). The experimental results are shown in Fig.3 (left). Since our basic RL algorithm is doing off-policy learning, it has

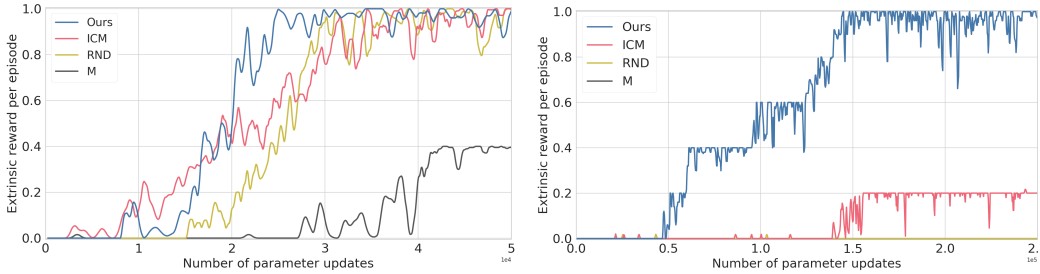

Figure 3: Extrinsic rewards per episode obtained in MyWayHome (left) and FlytrapEscape (right). Each plot shows the mean over 5 non-tuned random seeds. Figures showing the learning curves for each run can be found in the Appendix in Figure 14 and 13.

relatively decent random exploration capabilities. We see that the M agent is able to solve the task sometimes without any intrinsically generated motivations, but that all intrinsic motivation types help to solve the task more reliably and speed up the learning. Our method solve the task the fastest, but also ICM and RND learn to reach the goal reliably and efficiently.

We wanted to test the agents on a more difficult VizDoom map where structured exploration would be of vital importance. We thus designed a new map which scales up the navigation task of MyWay-Home. Inspired by how flytraps catch insects, we design the layout of the rooms in a geometrically challenging way that escaping from one room to the next with random actions is extremely unlikely. We show the layout of MyWayHome (Fig.2a) and FlytrapEscape (Fig.2b) with the same downscaling ratio. The maze consists of 4 rooms separated by V-shaped walls pointing inwards the rooms. The small exits of each room is located at the junction of the V-shape, which is extremely difficult to maneuver into without a sequence of precise movements. As in the MyWayHome task, in each episode, the agent starts from the red dot shown in Fig.2b with a random orientation. An episode terminates if the final goal is reached and the agent will receive a reward of $+1$, or if a maximum episode steps of $10,000$ ($2100$ for MyWayHome) is reached. The task is to escape the fourth room.

The experimental results on FlytrapEscape are shown in Fig.3 (right). Neither M nor RND manages to learn any useful policies. ICM solves the task in sometimes, while we can clearly observe that our method efficiently explores the map and reliably learns how to navigate to the goal. We visualize the learned successor features in Appendix E and its evolution over time is shown in the video `https://gofile.io/?c=HpEwTd`.

## 4.2 DEEPMIND LAB: EXPLORATION UNDER DISTRACTION

In the second experiment, we set out to evaluate if the agents would be able to reliably collect the faraway big reward in the presence of small nearby distractive rewards. For this experiment we use the 3D visual navigation simulator of DeepMind Lab (Beattie et al., 2016). We constructed a challenging level "AppleDistractions" (Fig.10b) with a maximum episode length of $1350$. In this level, the agent starts in the middle of the map (blue square) and can follow either of the two corridors. Each corridor has multiple sections and each section consists of two dead-ends and an entry to next section. Each section has different randomly generated floor and wall textures. One of the corridors (left) gives a small reward of $0.05$ for each apple collected, while the other one (right) contains a single big reward of $1$ at the end of its last section. The optimal policy would be to go for the single faraway big reward. But since the small apple rewards are much closer to the spawning location of the agent, the challenge here is to still explore other areas sufficiently often so that the optimal solution could be recovered.

The results are presented in Fig.4 (left). Ours received on average the highest rewards and is the only method that learns to navigate to the large reward in every run. The baseline methods get easily distracted by the small short-term rewards and do not reliably learn to navigate away from the distractions. With a separate policy for intrinsic motivation the agent can for some time interval completely "forget" about the extrinsic reward and purely explore, since it does not get distracted by the easily reachable apple rewards and can efficiently learn to explore the whole map. In the meanwhile the extrinsic policy can simultaneously learn from the new experiences and might learn about

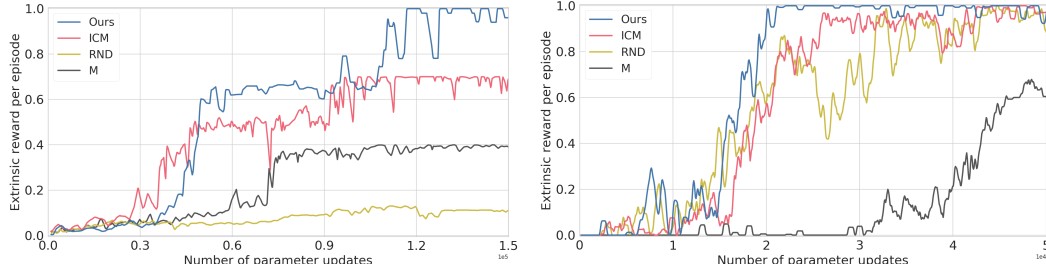

Figure 4: Extrinsic rewards per episode obtained in AppleDistractions (left) and Cartpole (right). Each plot shows the mean over 5 non-tuned random seeds. Left: Each agent is evaluated on the same 5 sets of random floor and wall textures, with 5 non-tuned environment seeds. In the ablation study (Appendix A) the SID variant outperforms the reward bonus variant of each of the 3 types of intrinsic rewards. Right: Ours also outperforms all baseline agents in the very different domain of classic control from pixels, which shows the general applicability of our proposed agent. Figures showing the learning curves for each run can be found in the Appendix in Figure 15 and 16.

the final goal discovered by the exploration policy. This highlights a big advantage of scheduling over bonus rewards, that it reduces the probability of converging to bad local optimums. In Section 4.4 we further showed that SID is generally applicable and also helps ICM and RND in this task.

## 4.3 DEEPMIND CONTROL SUITE: CLASSIC CONTROL FROM PIXELS

To show that our methods can be used in domains other than first-person visual navigation, we evaluate on the classic control task "cartpole: swingup_sparse" (DeepMind Control Suite Tassa et al. (2018)), using third-person view images as inputs (Fig.11). The pole starts pointing down and the agent receives a single terminal reward of $+1$ for swinging up the unactuated pole using only horizontal forces on the cart. Additional details are presented in Appendix D.3. The results are shown in Fig.4 (right). Compared to the previous tasks, this task is easy enough to be solved without intrinsic motivation, but we can see also that all intrinsic motivation methods significantly reduce the interaction time. Ours still outperforms other agents even in the absence of clear bottlenecks which shows its general applicability, but since the task is relatively less challenging for exploration, the performance gain is not as substantial as the previous experiments.

## 4.4 ABLATION STUDY

Further, we conducted an ablation study on AppleDistractions. We denote with "M+SFC", "M+RND", "M+ICM" the agents with one policy where the respective intrinsic reward is added to the extrinsic one. With "SID(M,SFC)", "SID(M,RND)", "SID(M,ICM)" the agents are named that have two policies, one for the respective intrinsic and one for the extrinsic reward and use SID

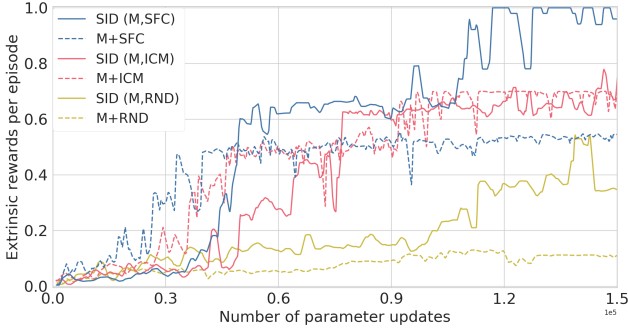

Figure 5: Ablation study results for AppleDistractions. Each plot shows the mean over 5 non-tuned random seeds.

to schedule between them. We note that the "SID(M,ICM)" agent corresponds to the "Ours" agent from the previous experiments. The results are presented in Figure 5. Our SID(M, SFC) agent received on average the highest rewards. Furthermore, we see that scheduling helped both ICM and SFC to find the goal and not settle for the small rewards, and SID also helps improve the performance of RND. The respective reward bonus counterparts of the three SID agents were more attracted to the small nearby rewards. This behavior is expected: By scheduling, the intrinsic policy of the SID agent is assigned with its own interaction time with the environment, during which it could completely "forget" about the extrinsic rewards. The agent then has a much higher probability of discovering the faraway big reward, thus escaping the distractions of the nearby small rewards. Once the intrinsic policy collects these experiences of the big reward, the extrinsic policy can immediately learn from those since both policies share the same replay buffer. Ablations for the other environments are reported in Appendix A.

## 5 CONCLUSION

In this paper, we investigate an alternative way of utilizing intrinsic motivation for exploration in DRL. We propose a hierarchical agent SID that schedules between following extrinsic and intrinsic drives. Moreover, we propose a new type of intrinsic reward SFC that is general and evaluates the intrinsic motivation based on longer time horizons. We conduct experiments in three sets of environments and show that both our contributions SID and SFC help greatly in improving exploration efficiency.

We consider many possible research directions that could stem from this work, including designing more efficient scheduling strategies, incorporating several intrinsic drives (that are possibly orthogonal and complementary) instead of only one into SID, testing our framework in other control domains such as manipulation, combining the successor representation with learned feature representations and extending our evaluation onto real robotics systems.

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

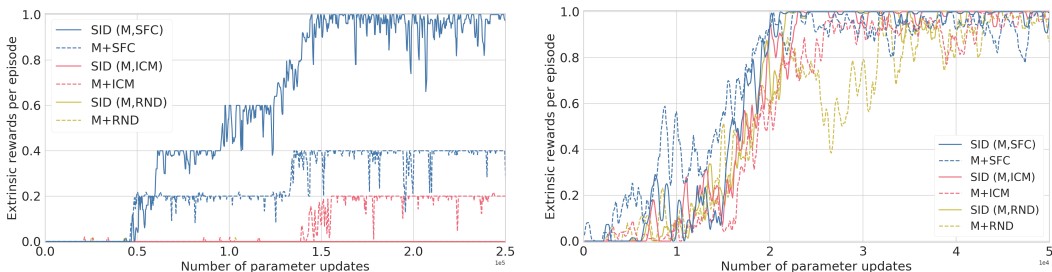

Figure 6: Ablation study results for FlytrapEscape (left) and Cartpole (right).

## A  APPENDIX: FURTHER ABLATION STUDIES

We have conducted ablation studies for all the three sets of environments to investigate the influence of scheduling on our proposed method, whether other reward types can benefit from scheduling too, and whether environment specific differences exist.

We compare the performance of the following agent configurations:

- Three reward bonus agents **M+ICM**, **M+RND**, **M+SFC**:
  The agent receives the intrinsic reward of ICM (Pathak et al., 2017), RND (Burda et al., 2018b) or our proposed SFC respectively as added bonus to the extrinsic main task reward and trains a mixture policy on this combined reward signal. We note that the **M+ICM** and **M+RND** agent in this section corresponds to the **ICM** and **RND** agent in all other sections respectively.

- Three SID agents **SID (M, ICM)**, **SID (M, RND)**, **SID (M, SFC)**:
  The agent schedules between following the extrinsic main task policy and the intrinsic policy trained with the ICM, RND or our proposed SFC reward respectively.

We note that the **SID (M, SFC)** agent in this section corresponds to the **Ours** agent in all other sections.

The results on AppleDistractions were shown in in Section 4.4 the main paper. In Fig.6 (left), we present the ablation study results for FlytrapEscape. The agents with the ICM component perform poorly. Only 1 run of M+ICM learned to navigate to the goal, while the scheduling agent SID(M,ICM) did not solve the task even once. But for the two SFC agents, the scheduling greatly improves the performance. Although the reward bonus agent M+SFC was not successful in every run, the SID(M,SFC) agent solved the FlytrapEscape in 5 out of 5 runs. We hypothesize the reason for the superior performance of SID(M,SFC) compared to M+SFC could be the following: Before seeing the final goal for the first time, the M+SFC agent is essentially learning purely on the SFC reward, which is equivalent to the intrinsic policy of the scheduling SID(M,SFC) agent. Since SFC might preferably go to bottleneck states as the difference between the SF of the two neighboring states are expected to be relatively larger for those states. Since the extrinsic policy is doing random exploration before receiving any reward signal, it could be a good candidate to explore the next new room from the current bottleneck state onwards. Then the SFs of the new room will be learned when it is being explored, which would then guide the agent to the next bottleneck regions. Thus the SID(M,SFC) agent could efficiently explore from bottleneck to bottleneck, while the M+SFC agent could not be able to benefit from the two different behaviors under the extrinsic and intrinsic rewards and could oscillate around bottleneck states. On the other hand, scheduling did not help ICM or RND. A reason could be that ICM or RND is not especially attracted by bottleneck states so it does not help exploration if the agent spends half of the time acting randomly as the extrinsic policy had no reward yet to learn from. Also since the FlytrapEscape environment is extremely exploration-challenging, the temporally extended view of our proposed SFC might of vital importance to guide efficient exploration.

In Fig.6 (right), we present the ablation study results for Cartpole. We can observe that SID helps to improve the performance of both ICM and RND. As for SFC, although the reward bonus agent

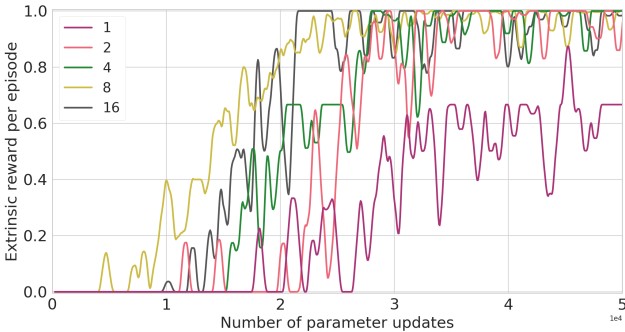

Figure 7: Results on "DoomMyWayHome" for different times of scheduling per episode in SID with SFC as intrinsic reward.

learns a bit faster than the SID agent, we note that actually all the three SID agent converge to more stable policies, while the reward bonus agents tend to oscillate around the optimal return.

## B    Appendix: Number of Switches per Episode

In experiments of the main paper with SID the scheduler chooses 8 times per episode which policy to follow until a new policy is chosen. We conducted a further experiment to examine how different numbers of switches per episode affects the performance of the agent. We carried out the experiment on "DoomMyWayHome" with an agent using SID with SFC for the intrinsic reward. The results are show in 7

## C    Appendix: Implementation Details

This section describes implementation details and design choices. The backbone of our algorithm implementation is presented in Section 3.4 and visualized in Fig. 8.

### C.1    Ape-X DQN

Since our algorithm requires an off-policy learning strategy, and in consideration for faster learning and less computation overhead, we use the state-of-the-art off-policy algorithm Ape-X DQN Horgan et al. (2018) with the $K$-step target ($K = 5$) for bootstraping without off-policy correction

$$y_t = \sum_{k=1}^{k=K} \gamma^{k-1} R_{t+k} + \gamma^K \max q(s_{t+K}, \arg\max_{a'} q(s_{t+K}, a'; \theta^-); \theta),$$

where $\theta^-$ denotes the target network parameters.

We chose the number of actors the be the highest the hardware supported, which was 8. To adapt the $\epsilon$ settings from the 360 actors in the Ape-X DQN to our setting of $N = 8$ actors, we set a fixed $\epsilon_i$ for each actor $i \in \{1, \ldots, 8\}$ as

$$\epsilon_i = \epsilon^{1 + \frac{(i-1)\frac{360}{N}}{360-1}\alpha}, \tag{5}$$

where $\alpha = 7$ and $\epsilon = 0.4$ are set as in the original work.

### C.2    Prioritized Experience Replay

For computational efficiency, we implement our own version of the prioritized experience replay. We split the replay buffer into two, with size of $40,000$ and $10,000$. Every transition is pushed to the first one, while in the second one only transitions are pushed on which a very large TD-error is computed. We store a running estimate of the mean and the standard deviation of the TD-errors

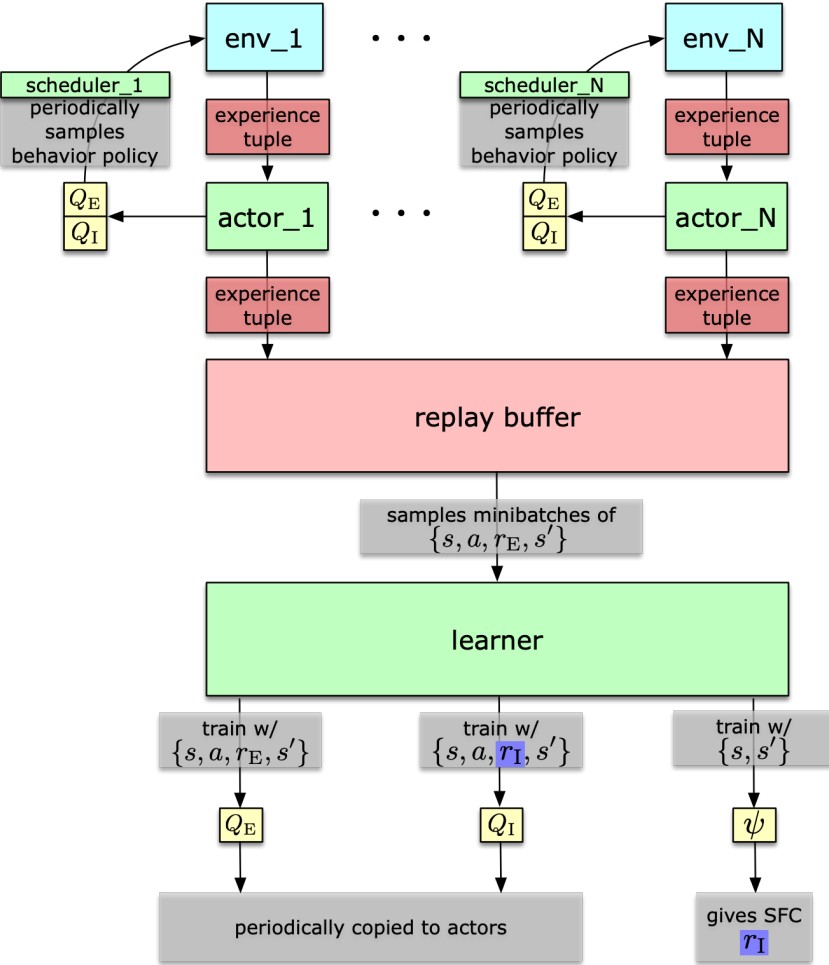

Figure 8: Flow diagram of the algorithm implementation (Sec.3.4).

and if for a transition the error is larger than the mean plus two times the standard deviation, the transition is pushed. In the learner a batch of size 128 consists of 96 transitions drawn from the normal replay buffer and 32 are drawn from the one that stores transition with high TD-error, which as a result have relatively seen a higher chance of being picked.

### C.3 SUCCESSOR FEATURE LEARNING

We note that previous works for learning the deep SF have included an auxiliary task of reconstruction on the features $\phi$ Kulkarni et al. (2016a); Zhang et al. (2017), while in this work we investigate learning $\psi$ without this extra reconstruction stream. Instead of adapting the features $\phi$ while learning the successor features $\psi$, we fix the randomly initialized $\phi$. This design follows the intuition that since SF ($\psi$) estimates the expectation of features ($\phi$) under the transition dynamics and the policy being followed, more stable learning of the SF could be achieved if the features are kept fixed.

The SF are learned from the same replay buffer as for training the Q-Net. Since our base algorithm is $K$-step Ape-X, and we follow the memory efficient implementation of the replay buffer as suggested in Ape-X, we only have access to $K$-step experience tuples ($K = 5$) for learning the SF. Therefore we calculate the intrinsic reward by applying the canonical extension of the SFC reward formulation (Eq.4) to $K$-step transitions

$$R_{t+K}^{\text{sfc}} = \|\psi_{\pi,\phi}(S_{t+K}) - \psi_{\pi,\phi}(S_t)\|_2^2. \tag{6}$$

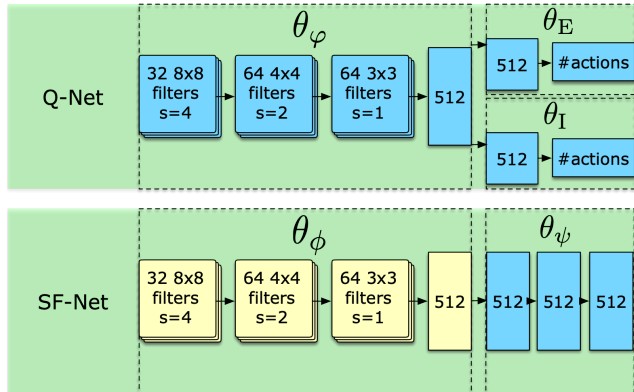

Figure 9: Model architecture for the **SID (M, SFC)** agent. Components with color yellow are randomly intialized and not trained during learning.

The behaviour policy $\pi$ associated with the SF is not given explicitly, but since the SF are learned from the replay buffer via TD learning, it is a mixture of current and past behaviour policies from all actors.

## C.4 REWARD NORMALIZATION

Most network parameters are shared for estimating the expected discounted return of the intrinsic and extrinsic rewards. The scale of the rewards has a big influence on the scale of the gradients for the network parameters. Hence, it is important that the rewards are roughly on the same scale, otherwise effectively different learning rates are applied. The loss of the network comes from the regression on the Q-values, which approximate the expected return. So our normalization method aims to bring the discounted return of both tasks into the same range. To do so we first normalize the intrinsic rewards by dividing them by a running estimate of their standard deviation. We also keep a running estimate of the mean of this normalized reward and denote it $r'_I$. Since every time step an intrinsic reward is received we estimate the discounted return via the geometric series. We scale the extrinsic task reward that is always in $\{0, 1\}$ with $\eta \frac{r'_I}{1-\gamma_I}$, where $\gamma_I$ is the discount rate for the intrinsic reward. Furthermore, $\eta$ is a hyperparameter which takes into account that for Q-values from states more distant to the goal the reward is discounted with the discount rate for the extrinsic reward depending on how far away that state is. In our experiments we set $\eta = 3$.

We did the same search for hyperparameters and normalization technique for all algorithms that include an intrinsic reward and found out that the procedure above works best for all of them. The algorithms were evaluated on the FlytrapEscape. For $\eta$ we tried the values in $\{0.3, 1, 3, 10\}$. We also tried to not normalize the rewards and just scale the intrinsic reward. To scale the intrinsic reward we tried the values $\{0.001, 0.01, 0.1, 1\}$. However, we found that as the scale of the intrinsic rewards is not the same over the whole training process this approach does not work well. We also tried to normalize the intrinsic rewards by dividing it by a running estimate of its standard deviation and then scale this quantity with a value in $\{0.01, 0.1, 1\}$.

## C.5 MODEL ARCHITECTURE

We use the same model architecture as depicted in Fig. 9 across all 3 sets of experiments.

ReLU activations are added after every layer except for the last layers of each dashed blocks in the above figure. For the experiments with the ICM (Pathak et al., 2017), we added BatchNorm (Ioffe & Szegedy, 2015) before activation for the embedding of the ICM module following the original code released by the authors. Code is implemented in pytorch (Paszke et al., 2017).

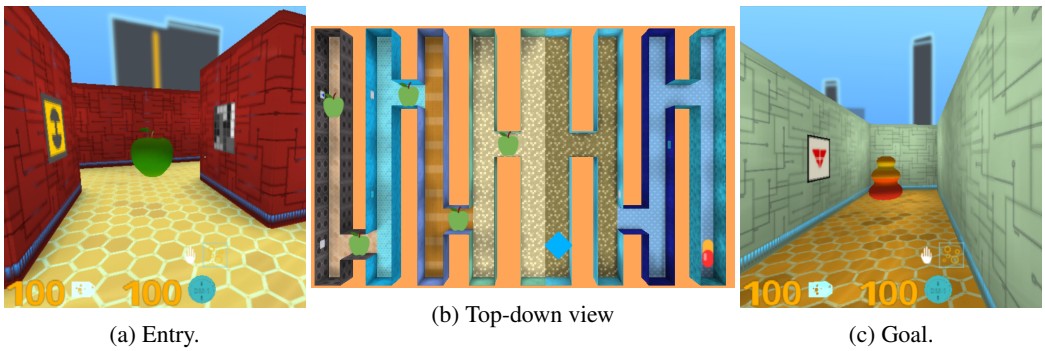

(a) Entry.  (b) Top-down view  (c) Goal.

Figure 10: Top-down view and exemplary first-person view observations captured in the AppleDistractions environment.

## D   APPENDIX: TRAINING DETAILS

We use a batch size of $128$ for all experiments the Adam optimizer (Kingma & Ba, 2014) with a learning rate of $1e - 4$.

For all experiments we used a stack of 4 consecutive, preprocessed observations as states.

For the first-person view experiments in VizDoom and DeepMind Lab, we use an action repetition of $4$, while for the classic control experiment we did not apply action repetition. In the text, we only refer to the actual environment steps (e.g. before divided by 4).

### D.1   ENVIRONMENT SETTINGS: VIZDOOM

The VizDoom environment produces $320 \times 240$ RGB images as observations. In a preprocessing step, we downscaled the images to $84 \times 84$ pixels and converted them to grayscale.

For FlytrapEscape, we adopted the action space settings from the MyWayHome task. The action space was given by the following 5 actions: *TURN_LEFT, TURN_RIGHT, MOVE_FORWARD, MOVE_LEFT, MOVE_RIGHT*

### D.2   ENVIRONMENT SETTINGS: DEEPMIND LAB

We setup the DmLab environment to produce $84 \times 84$ RGB images as observations. In Fig.10 we show examplary observations of AppleDistractions. We preprocessed the images by converting the observations to grayscale.

For a given enviroment seed, textures for each segment of the maze are generated at random.

We used the predefined DmLab actions from Espeholt et al. (2018). The action space was given by the following 8 actions (no shooting setting): *Forward, Backward, Strafe Left, Strafe Right, Look Left, Look Right, Forward+Look Left, Forward+Look Right*.

### D.3   ENVIRONMENT SETTINGS: DEEPMIND CONTROL SUITE

We conducted the experiments for the classic control task on the 'Cart-pole' domain with the 'swingup_sparse' task provided by the DeepMind Control Suite. Since our agents needs a discrete action space and the control suite only provides continuous action spaces, we discretized the single action dimension. The set of actions was {-0.5, -0.25, 0, 0.25, 0.5}. We configured the environment to produce $84 \times 84$ RGB pixel-only observations from the 1st camera, which is the only predefined camera that shows the full cart and pole at all times. We further convert the images to grey-scale and stack four consecutive frames as input to our network. The episode length was 200 environment steps.

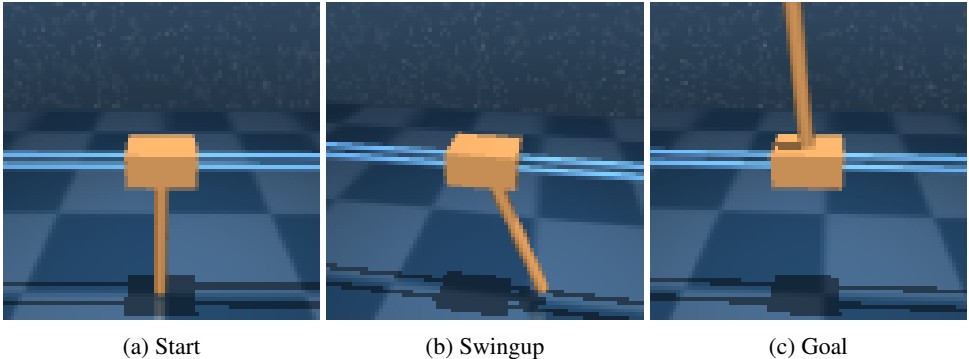

(a) Start            (b) Swingup            (c) Goal

Figure 11: Exemplary observations captured in the Cartpole environment.

### D.4 ENVIRONMENT SETTINGS: FOUR-ROOM

In the four-room domain, the agent can transition to directly connected states using the four actions
'up', 'down', 'left' and 'right'. For Fig.1 a),b) the successor features were calculated analytically
via the formula $\mathbf{\Psi} = (\mathbf{I} - \gamma\mathbf{P})^{-}1$, where $P$ denotes the one-step transition matrix. The SF discount
factor was set to $\gamma = 0.95$.

For Fig.1 c),d) the agents performed 10000 episodes with 30 steps each. These short episodes ensure
that exploration remains challenging, even in a relatively small environment. In d), the learning rate
for the SF as well as the Q-table was set to $0.05$. To prevent optimistic initialization effects the
Q-table was initialized to $0$.

### D.5 INFRASTRUCTURE

To generate our results we used two machines that run Ubuntu 16.04. Each machine has 4 GeForce
Titan X (Pascal) GPUs. On one machine we run 4 experiments in parallel, each experiment on a
separate GPU.

## E APPENDIX: SUCCESSOR DISTANCE VISUALIZATION

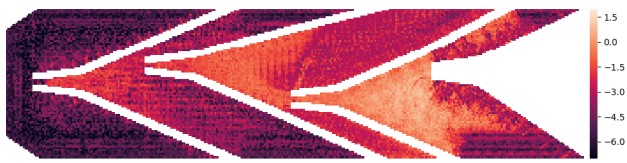

Figure 12: Projection of the SFs. For the purpose of visualization we discretized the map into
$85 \times 330$ grids and position the trained agent SID(M,SFC) at each grid, then computed the successor
features $\psi$ for that location for each of the 4 orientations $(0°, 90°, 180°, 270°)$, which resulted in a
$4 \times 512$ matrix. We then calculated the $l2$-difference of this matrix with a $4 \times 512$ vector containing
the successor features of the starting position with the 4 different orientations. Shown in $\log$-scale.

As an additional evaluation, we visualize the SF of an Ours agent (i.e. trained with SID which
schedules between the extrinsic policy and the SFC policy) at the end of the training (Fig.12). That
means the SF are trained with the experiences from the behaviour policy of our agent. We can see
that the SD from each coordinate to the starting position tends to grow as the geometric distance
increases, especially for those that locate on the pathways leading to later rooms. This shows that
the learned SD and the geometric distance are in good agreement and that the SF are learned as
expected. Furthermore, we observe big intensity changes around the bottlenecks (the room entries)
in the heatmap, which also supports the hypothesis that SFC leads the agent to bottleneck states.
We believe this is the first time that SF are shown to behave in a first-person view environment

as one would expect from its definition. The evolution of the SF over time is shown in the video `https://gofile.io/?c=HpEwTd`.

# F  APPENDIX: MORE DETAILS ON THE SCHEDULER

When learning optimal value functions or optimal policies via TD or policy gradient with deep function approximators, optimizing with algorithms such as gradient descent means that the policy would only evolve incrementally: It is necessary that the TD-target values do not change drastically over a short period of time in order for the gradient updates to be meaningful. The common practice of utilizing a target network in off-policy DRL (Mnih et al., 2015) stabilizes the update but in the meanwhile making the policy adapt even more incrementally over each step.

But intrinsically motivated exploration, or exploration in general, might benefit from an opposite treatment of the policy update. This is because the intrinsic reward is non-stationary by nature, as well as the fact that the exploration policy should reflect the optimal strategy corresponding to the current stage of learning, and thus is also non-stationary.

With the commonly adopted way of using intrinsic reward as a bonus to the extrinsic reward and train a mixture policy on top, exploration would be a balancing act between the incrementally updated target values for stable learning and the dynamically adapted intrinsic signals for efficient exploration. Moreover, neither the extrinsic nor the intrinsic signal is followed for an extended amout of time.

Therefore, we propose to address this issue with a hierarchical approach that by design has slowly changing target values while still allowing drastic behavior changes. The idea is to learn not a single, but multiple policies, with each one optimizing on a different reward function. To be more specific, we assume to have $N$ tasks $\mathbb{T} \in \mathcal{T}$ (e.g. $N = 2$ and $\mathcal{T} = \{\mathbb{T}_E, \mathbb{T}_I\}$ where $\mathbb{T}_E$ denotes the extrinsic task and $\mathbb{T}_I$ the intrinsic task) defined by $N$ reward functions (e.g. $R_E$ and $R_I$) that share the state and action space. The optimal policy for each of these $N$ different MDPs can be learned with arbitrary off-policy DRL algorithms. During each episode, a high-level scheduler periodically selects a policy for the agent to follow to gather experiences, and each policy is trained with all experiences collected following those $N$ different policies. The overall learning objective is to maximize the extrinsic reward $\mathbb{E}_{\omega(\mathbb{T}|S_t)}\mathbb{E}_{\pi_\mathbb{T}(A_t|S_t)}\left[q_{\mathbb{T}_E}(S_t, A_t|A_t \sim \pi_\mathbb{T}(\cdot|S_t))\right]$ ($\omega$: the macro-policy of the scheduler).

By allowing the agent to follow one motivation at a time, it is possible to have a pool of $N$ different behavior policies without creating unstable targets for off-policy learning. By scheduling $M$ times even during an episode, we implicitly increase the behavior policy space by exponential to $N^M$ for a single episode. Moreover, disentangling the extrinsic and intrinsic policy strictly separates stationary and non-stationary behaviors, and the different sub-objectives would each be allocated with its own interaction time, such that extrinsic reward maximization and exploration do not distract each other. We investigated several types of high-level schedulers, however, none of them consistently outperforms a random one. We suspect the reason why a random scheduler already performs very well under the SID framework, is that a highly stochastic schedule can be beneficial to make full use of the big behavior policy space.

We investigated three types of high-level schedulers:

- Random scheduler: Sample a task from uniform distribution every task steps.
- Switching scheduler: Sequentially switches between extrinsic and intrinsic task.
- Macro-Q Scheduler: Learn a scheduler that learns with macro actions and from subsampled experience tuples. In each actor, we keep an additional local buffer that stores $N + 1$ subsampled experiences: $\{s_{t-Nm}, \ldots, s_{t-2m}, s_{t-m}, s_t\}$. Then at each environment step, Besides the $K$-step experience tuple mentioned above, we also store an additional macro-transition $\{s_{t-Nm}, s_t\}$ along with its sum of discounted rewards to the shared replay buffer. This macro-transition is paired with the current task as its macro-action. The Macro-Q Scheduler is then learned with an additional output head attached to $\theta_\varphi$ (we also tried $\theta_\phi$).
- Threshold-Q Scheduler: Selects task according to the Q-value output of the extrinsic task head. For this scheduler no additional learning is needed. It just selects a task based on the current Q-value of the extrinsic head $\theta_e$. We tried the following selection strategies:

- Running mean: select intrinsic when the current Q-value of the extrinsic head is below its running mean, extrinsic otherwise
- Heuristic median: observing that the running mean of the Q-values might not be a good statistics for selecting tasks due to the very unevenly distributed Q-values across the map, we choose a fixed value that is around the median of the Q-values (0.007), and choose intrinsic when below, extrinsic otherwise

As we report in the paper, none of the above scheduler choices consistently performs better across all environments than a random scheduler. We leave this part to future work.

## G APPENDIX: SINGLE RUNS

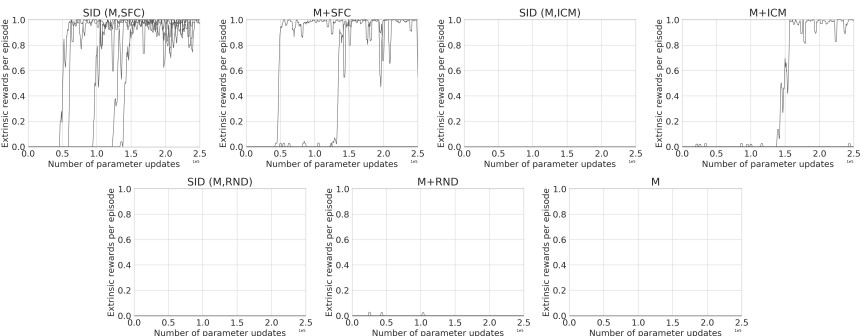

Figure 13: Learning curves for each run for each agent trained on the flytrap environment.

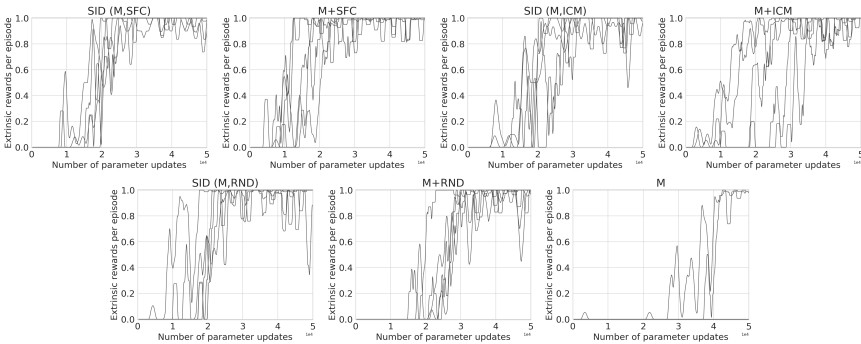

Figure 14: Learning curves for each run for each agent trained on the MyWayHome environment.

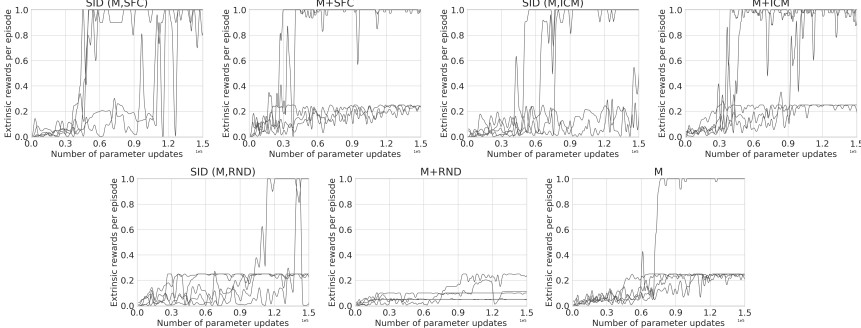

Figure 15: Learning curves for each run for each agent trained on the AppleDistraction environment.

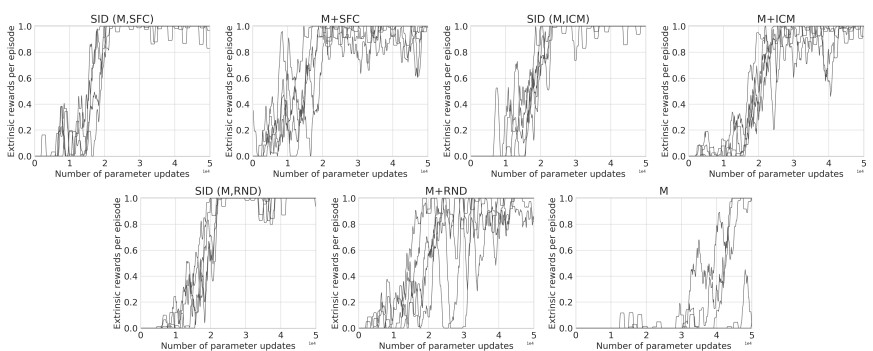

Figure 16: Learning curves for each run for each agent trained on the Cartpole environment.

