# OpenReview forum: "Scheduled Intrinsic Drive: A Hierarchical Take on Intrinsically Motivated Exploration"
_ICLR.cc/2020/Conference — Reject_

### Official Review · AnonReviewer2 · 2019-10-23
**Official Blind Review #2**

**Rating:** 3

**Review:**

Summary:

This paper proposes the use of a controller that selects whether to act according to a policy trained to maximize an intrinsic reward or a different policy trained to maximize the extrinsic reward of a task. The two policies are trained jointly and off-policy. However, the controller is not trained for their experiments and instead randomly (with equal probability) picks one of the two policies every N steps (with N fixed). They also introduce a new kind of . intrinsic reward based on successor features, that is supposed to capture trajectory statistics for a fixed policy. They name their method scheduled intrinsic drive (SID).

Main Comments:

While this paper proposes some interesting ideas,  I am concerned about the soundness of the method, some of the precise implementation details and I believe the empirical evaluation could be greatly improved.

One of my main concerns is the soundness of using the SFC as intrinsic reward while training off-policy. SFs are defined for a fixed policy so they capture statistics of future states if that policy is being used for control. Can you provide more explanation for why the SFC should still be a useful signal for the agent in the case in which it will follow a very different policy (which seems likely given that the replay buffer not only contains a mix of the exploration and exploitation policies, but also policies at different points during training with potentially very different state visitation distributions). Are the SFs trained with data from both the exploration and the exploitation policy? How can we expect the SFC to have useful signal since it is trained using such a wide range of policies?

Is there any guarantee that the arbitrary feature embeddings (which are not learned in your experiments if I understood correctly) and thus the successor features (SFs) will contain meaningful information about the kinds of states a policy will visit in the future? An ablation using hand-designed feature embeddings that contain relevant information about the state (i.e. in a gridworld) might be useful to understand how it compares to a randomly initialized network, which is what you used for the state embeddings as I understand it.

I am also concerned by the novelty of this work and the fact that it is missing references and discussion to prior work that proposes very similar ideas. For example, [1] proposed the optimization of different losses at the same time: one for exploitation and one or more for exploration. Can you please discuss what is the difference between your method and theirs (other than the intrinsic reward used for the exploration policy)? Similarly, [2] attempts to decouple exploration and exploitation in RL. This reference (and perhaps others that I have missed) should be included and discussed in the paper.

The empirical validation is missing important statistics such as variance across runs. Experiments on AppleDistractions and Cartpole only have 3 random seeds which I do not think is enough for drawing conclusions confidently. Moreover, on the simpler and standard tasks, SID does not seem to be significantly better than other baselines. It is only on  carefully designed tasks (e.g. FlytrapEscape or AppleDistractions) that are not regularly used as benchmarks that the method seems to perform better.

The experiments section could be improved by including other (more powerful) baselines such as count/pseudocount exploration methods which have been shown to be more effective than ICM / RND for certain benchmarks, the paper using an intrinsic reward based on successor representations [3] or even Go-Explore [4] that is specifically designed to deal with distractor objects for the AppleDistractions task. Additionally, evaluating SID on harder exploration tasks that are generally considered to be good benchmarks by the community would be helpful (e.g. Montezuma Revenge, Pitfall, sparser versions of DoomMyWayHome etc.) would also strengthen the experimental section.

Other Questions / Comments:

1. There is no measure of the variance / standard deviation across the random seeds in any of the plots. I find it necessary to be included in the plots, along with the mean across runs.

2. What is the reasoning behind using the number of updates (instead of e.g. number of frames / steps / episodes) in the plots? How exactly do you measure the number of updates that appears in the plots? Is that the total number of updates used for the control policy, the exploration policy, and the successor features or is it only the number of updates used for the control policy?

3. I find the use of the term "hierarchical" in the title and throughout the paper to be misleading since this term is usually used with a different meaning in the RL literature (i.e. to refer to options/subpolicies that a higher-level policy might choose to pursue at a given time). In your case, the control policy is one of the subpolicies and the other subpolicy is only used for exploration.

4. The paper also contains claims which I find unsubstantiated by the results / analytical formulation such as: " our proposed SFC reward implicitly captures statistics over the full distribution of policies that have been followed,
since the successor features are learned using states sampled from all past experiences" on page 2 or "Another valuable property of SFC is that it adapts in very meaningful ways that lead to efficient
non-stationary exploration policies, when the transitions gathered by a policy maximizing the SFC
reward is used to update the SF itself" on page 5. Please provide more intuition or theoretical / empirical evidence to support such claims.

5. What do you use for the fixed interval (N) at which the meta-controller is choosing which policy to follow? Have you tried training the meta-controller? It would be interesting to see how the results change as N varies. Is N = 1 better than N = length of episode or the other way around or does the choice of N not matter that much?



References:
[1] Beyer, Lucas, et al. "MULEX: Disentangling Exploitation from Exploration in Deep RL." arXiv preprint arXiv:1907.00868 (2019).
[2] Cédric Colas, Olivier Sigaud, and Pierre-Yves Oudeyer. GEP-PG: Decoupling Exploration and
Exploitation in Deep Reinforcement Learning Algorithms. In Proceedings of the International
Conference on Machine Learning (ICML), 2018.
[3] Marlos C Machado, Clemens Rosenbaum, Xiaoxiao Guo, Miao Liu, Gerald Tesauro, and Murray
Campbell. Eigenoption discovery through the deep successor representation. arXiv preprint
arXiv:1710.11089, 2017.
[4] Ecoffet, Adrien, et al. "Go-explore: a new approach for hard-exploration problems." arXiv preprint arXiv:1901.10995 (2019).


**Experience Assessment:**

I have read many papers in this area.

**Review Assessment: Checking Correctness Of Derivations And Theory:**

I assessed the sensibility of the derivations and theory.

**Review Assessment: Checking Correctness Of Experiments:**

I carefully checked the experiments.

**Review Assessment: Thoroughness In Paper Reading:**

I read the paper thoroughly.

---

> ### Author Response · Authors · 2019-11-12
> **Response to Reviewer #2  (2/2)**
>
>
> $\bullet$ Evaluation on common benchmarks
>
> Response:
> We note that we evaluated our method on the very sparse version (the hardest level) of Mywayhome in our paper, where it outperformed all compared methods.
> To evaluate it on more diverse environments we conducted preliminary experiments on Montezumas Revenge. We are aware that many works proposed to solve this game, but we also speculate that this environment poses a special kind of exploration challenge, where one uncareful exploration move would make the agent die.
> We can not provide conclusive results on this environment given the limited time and computing resources. However, our preliminary results show that our method is general enough to also help in this environment, while we do not claim that our method achieves state-of-the-art results on Montezuma. A plot of the results can be found under https://gofile.io/?c=0XM4LG .
> Because of the time constraints we could only test our method using SFC and SID against a pure extrinsic reward agent. The base RL algorithm was again Ape-X with 8 actors with a replay memory of 500k, run for 400k gradient updates which corresponds to roughly 200 million frames.
>
> $\bullet$  Comment:
> “What is the reasoning behind using the number of updates (instead of e.g. number of frames / steps / episodes) in the plots? How exactly do you measure the number of updates that appears in the plots? Is that the total number of updates used for the control policy, the exploration policy, and the successor features or is it only the number of updates used for the control policy?”
>
> Response:
> The number of updates is the number of gradient steps. In one such step one gradient update is applied to all trained parts (the Q-net and the SF-net). We used the number of updates in the plots following prior work on intrinsic motivation (e.g. [5], [6]).
>
> $\bullet$ Comment:
> “There is no measure of the variance / standard deviation across the random seeds in any of the plots.“
>
> Response:
> We agree that usually the standard deviation should be plotted. However in the paper we choose to show the curves of each individual runs in the Appendix G instead of showing the std, since plotting the standart deviation makes our figures harder to interpret. The reason is that in a terminal reward setting each run results in a total reward of 0 or 1. Using the sample variance leads to unintuitive results such as overconfident error predictions if all or no runs converged or underconfident predictions which can span negative values, even though the environment just provides positive reward.
>
> $\bullet$ Unsubstantiated formulation
>
> Response:
> We agree that some formulations were misleading and have revised it (e.g. on page 2).
>
> $\bullet$  Comment:
> “Experiments on AppleDistractions and Cartpole only have 3 random seeds”
> Response:
> We apologize as this was actually a typo from us, we used 5 random seeds, as can also be seen in Figure 15 and 16 in Appendix G, where the learning curves of the single runs are plotted.
>
> $\bullet$ Comment:
> “What do you use for the fixed interval (N) at which the meta-controller is choosing which policy to follow? Have you tried training the meta-controller? It would be interesting to see how the results change as N varies. Is N = 1 better than N = length of episode or the other way around or does the choice of N not matter that much?”
>
> Response:
> We conducted a small experiment on my_way_home with different choices for the number of switches and added the results to the appendix (Appendix B). The results show that our method is not very sensitive about the exact number of switches, however switching just once per episode performed worse.
>
> [1] Beyer, Lucas, et al. "MULEX: Disentangling Exploitation from Exploration in Deep RL." arXiv preprint arXiv:1907.00868 (2019).
> [2] Cédric Colas, Olivier Sigaud, and Pierre-Yves Oudeyer. GEP-PG: Decoupling Exploration and Exploitation in Deep Reinforcement Learning Algorithms. In Proceedings of the International
> [3] Yuri Burda, Harri Edwards, Deepak Pathak, Amos Storkey, Trevor Darrell, and Alexei A Efros.Large-scale study of curiosity-driven learning.arXiv preprint arXiv:1808.04355, 2018a.
> [4] Martin Riedmiller, Roland Hafner, Thomas Lampe, Michael Neunert, Jonas Degrave, Tom Van deWiele,  Volodymyr Mnih,  Nicolas Heess,  and Jost Tobias Springenberg.   Learning by playing-solving sparse reward tasks from scratch.arXiv preprint arXiv:1802.10567, 2018
> [5] Yuri Burda, Harrison Edwards, Amos Storkey, and Oleg Klimov. Exploration by random networkdistillation.arXiv preprint arXiv:1810.12894, 2018b.
> [6] Deepak Pathak, Pulkit Agrawal, Alexei A. Efros, and Trevor Darrell. Curiosity-driven exploration by self-supervised prediction. InInternational Conference on Machine Learning (ICML), 2017.

---

> ### Author Response · Authors · 2019-11-12
> **Response to Reviewer #2 (1/2)**
>
> We thank the reviewer for the detailed comments and the effort spent on reviewing our paper. Below we address the concerns raised by the reviewer point by point.
>
> $\bullet$  Differences to [1],[2]
>
> Response:
> We thank the reviewers for the references. Below we discuss their differences from our paper.
> [1] proposes a similar hierarchical approach as ours, however, we note that our previous workshop paper (an anonymous version of our workshop paper can be found here: https://gofile.io/?c=iFfYT7 ) is concurrent with it. We added a citation to this work in the updated version.
> [2] differs from us in the approach and the application domain. Their decoupling of exploration and exploitation is implemented by a pure exploration stage followed by a pure exploitation stage. This differs fundamentally from our method to schedule different objectives (e.g. extrinsic and intrinsic drives) within an episode. Furthermore their method targets exploration challenges specifically brought by high dimensional continuous action space, while we focus on more general sparse exploration.
>
> $\bullet$  Fixed vs. learned feature embeddings
>
> Response:
> In our preliminary experiments we found that the performance of training the feature embedding $\theta_\phi$ is roughly comparable as keeping the randomly initialized features fixed. Our rationale for fixing the features is:
> As shown in [3], fixing randomly initialized features gives comparable results as with training features for ICM. From our experience with the commonly adopted way of training the SF with an autoencoder structure, we experienced that the interleave between training the two objectives (learning the features and rewards, learning the SF) is relatively sensitive to hyperparameter settings. Due to those reasons we wanted to investigate the possibility of minimizing the procedure and the computation budget required for learning the SF, just as [3] investigated the possibility of using a fixed feature extractor. As the experiments suggested, this setup can serve our method for the experiments considered, as although randomly initialized, the CNN structure is already a good prior for feature extractors at least in the visual input domain.
> We note that the visualization of the SFs which can be seen in Figure 12 in Appendix E and in the video show that the learned SFs are meaningful with fixed feature embedding.
>
> $\bullet$ Hierarchical framework
>
> Response:
> We designed and investigated different kinds of high-level schedulers, other than the random one, which we listed in Appendix E. We note that hierarchical refers to methods that involves a chained level of decision making, as in this paper we have a high-level scheduler and low-level sub-policies.
> We agree with the reviewer that our hierarchical setup is quite different from the previous options framework, where the optimal policy is constructed by a set of options.
> Other than the options framework, [4] proposes another line of hierarchical RL which inspired our work. While their method schedules between the extrinsic task and a set of pre-defined auxiliary tasks such as maximizing touch sensor readings, we adapted it to find an alternative solution to utilize intrinsic motivation other than the commonly adopted reward bonus setup.
> We note that our scheduler schedules several times (e.g. 8 times) during an episode. Since we have 2 sub-policies to choose from, the behavior policy space is implicitly increased exponentially to 2^8 which can greatly increase the behavior diversity and help exploration. We added more discussion of the scheduling choices in Appendix E.
> For the scheduler, as discussed in App. E, we chose the random one as empirically none of the other schedulers we tried consistently outperform it. By Occam’s razor we present the random one as in our final method. Instead of viewing it as a disadvantage, we kindly invite the reviewer to see it as an advantage, that the simplest scheduling is able to bring performance gain.
>
> $\bullet$ Training SFC off-policy
>
> Response:
> The reviewer is correct that the replay buffer stores experiences generated from the behavior policy, which is a mixture of exploration and exploitation experiences, from different points during training. The SF is trained using samples from this replay buffer, which contains a mixture of the agent's past trajectories. But we note that training SF using this mixture of experiences is not a technical flaw, but it is particularly intended to do so. When learning on this mixture of policies, the SF estimates the state visitation distributions of its past trajectories in the replay buffer. We also note older trajectories are continuously replaced by more recent ones. Therefore the influence of older behavior policies are gradually washed out from the SF. Then the learned SF will be a correct indicator of how often it has explored certain regions relatively recently. In that sense the SF are trained “on-policy” with respect to this mixture of policies.

---

### Official Review · AnonReviewer1 · 2019-10-26
**Official Blind Review #1**

**Rating:** 8

**Review:**

This paper tackles the problem of how to integrate intrinsic rewards most effectively, in the episodic sparse reward setting. It has two main technical contributions. The first is Scheduled Intrinsic Drive (SID), which trains two separate policies -- one for maximizing the extrinsic (i.e., task) reward and another for maximizing the intrinsic reward -- rather than a single policy that maximizes a weighted combination of both. This uses the same training setup as prior work, Scheduled Auxiliary Control (SAC), except here the extra policy is trained on intrinsic reward rather than an auxiliary task. The second contribution is Successor Feature Control (SFC), a novel approach for computing intrinsic rewards. For a given transition (s, a, s'), the intrinsic reward from SFC is the squared difference in successor features between states s and s'. Since successor features encompass a notion of which kinds of states the agent will encounter in the future after starting from the current state, this type of intrinsic reward is more far-sighted than most state-of-the-art approaches. Empirical analysis shows that SFC leads agents to explore bottleneck states, which is especially helpful for solving navigation tasks.

This paper is well-motivated and clearly written. The experimental evaluation of this paper is thorough, comparing SID to adding extrinsic and intrinsic reward together, and comparing SFC to two recent approaches for generating intrinsic rewards, ICM and RND. The appendix does a good job of providing implementation details for reproducibility, in particular regarding reward normalization and the variation of prioritized experience replay. I also greatly appreciate that design decisions are justified, for instance that the choice of using a random scheduler was made because it outperformed several versions of a learned scheduler.

My only concerns with the paper have to do with evaluation. SFC is compared to prior approaches for computing intrinsic rewards that only take into account transition-level information, whereas SFC takes into account trajectory-level information, and naturally performs better. But there are also recent approaches that do take into account trajectory-level information in different ways, e.g. Savinov et al. (2018). SFC should also be compared to approaches in this category.

I would also like to see an analysis of the failure cases that SFC is vulnerable to. Currently the evaluation domains used, with the exception of cartpole, are all tasks involving first-person navigation. So I wonder whether SFC is most effective (compared to existing approaches) on primarily these tasks in this domain, that are partially observable. It would be nice to see a wider variety of evaluation domains, for instance Montezuma's Revenge, which is frequently used to evaluate algorithms for computing intrinsic rewards, as well as other methods for improving exploration of RL agents. It would be neat if agents trained using SFC are better able to navigate through the doors in this game, since that seems to be a clear example of bottlenecks.

Minor questions / comments:
- In Figure 1b, why are the values on the four bottlenecks not all exactly the same? The maze is symmetric, so I would expect them to be equal.
- The plots in Figures 3 through 6 should show the standard deviation.

Typos:
- Page 2, "inexplicitely" --> "implicitly"
- Page 4, "temporarily" --> "temporally"
- Page 8, "carpole" —> "cartpole"

**Experience Assessment:**

I have published one or two papers in this area.

**Review Assessment: Checking Correctness Of Derivations And Theory:**

I carefully checked the derivations and theory.

**Review Assessment: Checking Correctness Of Experiments:**

I carefully checked the experiments.

**Review Assessment: Thoroughness In Paper Reading:**

I read the paper at least twice and used my best judgement in assessing the paper.

---

> ### Author Response · Authors · 2019-11-12
> **Response to Reviewer #1**
>
> We thank the reviewer the effort spent on reviewing our paper and the constructive remarks. We highly appreciated the positive feedback.
>
> $\bullet$ Comment:
> “Would be nice to see a wider variety of evaluation domains, for instance Montezuma's Revenge”
>
> Response:
> We conducted an additional experiment on Montezuma’s Revenge. While we think that it tests for a very specific form of exploration that does not necessarily generalizes to many other environments, we agree that the environment can be used to test the generality of our approach.
> A plot of the result can be found at https://gofile.io/?c=0XM4LG .
> Because of the limited time we can not give conclusive results and only tested our agent using SID and SFC against a pure extrinsic reward agent. We do not claim that our method achieves state-of-the-art results on Montezuma but the results demonstrate that our method also helps in an environment that is very different to the environments shown in the main paper.
> The base RL algorithm was again Ape-X with 8 actors with a replay memory of 500k, run for 400k gradient updates which corresponds roughly to 200 million frames.
>
> $\bullet$ Comparison to approaches that do take into account trajectory-level information
>
> Response:
> Actually we had conducted experiments using the intrinsic reward proposed in [1] which also generates the intrinsic reward signal based on the successor features. Unfortunately, our implementation of the method into our framework was outperformed by all other baselines, which is why we did not include it.
>
> $\bullet$ Comment:
> “In Figure 1b, why are the values on the four bottlenecks not all exactly the same? The maze is symmetric, so I would expect them to be equal.”
>
> Response:
> This is a very insightful and interesting point. The expectation of the reviewer is correct that the values for the bottlenecks would be exactly the same if the maze were symmetric. However as originally proposed in [2] the rooms are slightly imbalanced (the rooms on the left have 25 different states each, whereas the top right has 30 and the bottom right has 20).
> We observed that this asymmetry is reflected by the values of the bottleneck states. When comparing the SFC rewards of the states that lie close to the doors, we see that those states that lie in the bigger rooms have smaller SFC reward than those lie in smaller rooms. This agrees with the intuition that in smaller rooms there is less to explore, therefore leaving the smaller rooms corresponds to higher reward in comparison to leaving bigger rooms.
>
> $\bullet$ Comment:
> “The plots in Figures 3 through 6 should show the standard deviation.”
>
> Response:
> We agree that usually the standard deviation should be plotted. However in the paper we choose to show the curves of each individual runs in the Appendix G instead of showing the standard deviation, since plotting the standard deviation makes our figures harder to interpret. Previous works [3] also chose to plot each of the individual runs instead of showing the standard deviation.
>
> The reason is that in a terminal reward setting each run results in a total reward of 0 or 1. Using the sample variance leads to unintuitive results such as overconfident error predictions if all or no runs converged or underconfident predictions which can span negative values, even though the environment just provides positive reward.
>
>
> [1] Marlos C Machado, Marc G Bellemare, and Michael Bowling. Count-based exploration with the successor representation. arXiv preprint arXiv:1807.11622, 2018.
> [2] Richard S Sutton, Doina Precup, and Satinder Singh. Between mdps and semi-mdps: A framework for temporal abstraction in reinforcement learning. Artificial intelligence, 112(1-2):181– 211, 1999.
> [3] Nikolay Savinov, Anton Raichuk, Rapha ̈el Marinier, Damien Vincent, Marc Pollefeys, Timo-thy Lillicrap, and Sylvain Gelly.Episodic curiosity through reachability.arXiv preprintarXiv:1810.02274, 2018.

---

### Official Review · AnonReviewer3 · 2019-11-03
**Official Blind Review #3**

**Rating:** 6

**Review:**

Summary: A very nice study on the benefits of successor feature control as an intrinsic drive for hard exploration problems. The work builds nicely on previous work on using SF for exploration and proposes using derived (reachability under $\psi^{\pi}$) distances  as intrinsic motivation for an (purely) exploratory policy. This exploratory strategy will be used in conjunction with a policy trained on the extrinsic reward to gather (off-policy) data for both learning processes. The author proposed 'combining' these two policies via a simple scheduler, similar to (Riedmiller 2018).

Good paper/addition to both the intrinsic motivation literature and SFs studies.

Positives:
1) I like the separation of concerns achieved by training separate policies train for the two reward signals (intrinsic and extrinsic).
2) SFC as intrinsic reward and the study comparing this with other intrinsic signals (ICM/RND). The more interesting study might in the appendix though (Appendix A). I would suggest moving that into the main paper, as it nice separate the influence of scheduling component and the 'quality' of the proposed intrinsic reward.
3) Carefully conducted study, with relevant ($\epsilon$-SOTA) baselines and ablation studies.

Points of improvement or clarification:
1) The SFs and the derived reward were done based on random pseudo-rewards $\phi$ (Pg 5, SF-Nets). It maybe worth exploring learning those to capture more interesting features of the task at hand, especially in situation were there is more signal in the extrinsic reward. Do the authors have a sense of how problematic changing this component throughout training would be? As this acts are a reward signal to the inference the intrinsic reward signal, which then trains the exploratory policy. Thus small changes in one, can have massive implications for the trained Q-net, $Q_{E}$.
2) It wasn't clear from the exposition which policy is used to train the SFs? The exploratory policy, the uniform random one or the behaviour policy (the combination between the two trained policies given by the Q-nets).
3) On the SID setup. Did you conduct any studies on M (the number of switches)? For instance, how does this compare with something like episode switching, which has been explored before?
4) There are a couple of observation/discussion claims that are not really substantiated (for instance, last paragraph in Sec. 3.2). The paper is fine content-wise, without them. I would strongly suggest either removing them, re-phasing them as hypothesis and/or back them by more evidence.
5) The link to the video (https://gofile.io/?c=HpEwTd.) doesn't work. Please update.




**Experience Assessment:**

I have published in this field for several years.

**Review Assessment: Checking Correctness Of Derivations And Theory:**

I assessed the sensibility of the derivations and theory.

**Review Assessment: Checking Correctness Of Experiments:**

I assessed the sensibility of the experiments.

**Review Assessment: Thoroughness In Paper Reading:**

I read the paper at least twice and used my best judgement in assessing the paper.

---

> ### Author Response · Authors · 2019-11-12
> **Response to Reviewer #3**
>
> We thank the reviewer for the insightful comments and suggestions, we highly appreciate the positive feedback.
>
> $\bullet$ Comment:
> “SFC as intrinsic reward and the study comparing this with other intrinsic signals (ICM/RND). The more interesting study might in the appendix though (Appendix A). I would suggest moving that into the main paper, as it nice separate the influence of scheduling component and the 'quality' of the proposed intrinsic reward.”
>
> Response:
> As the reviewer suggested we have incorporated some of the contents of Appendix A to the main paper in the revised version.
>
> $\bullet$ Comment:
> “The SFs and the derived reward were done based on random pseudo-rewards (Pg 5, SF-Nets). It maybe worth exploring learning those to capture more interesting features of the task at hand, especially in situations were there is more signal in the extrinsic reward. Do the authors have a sense of how problematic changing this component throughout training would be?”
>
> Response:
> We thank the reviewer for this comment.
> In our preliminary experiments we found that the performance of training the feature embedding $\theta_\phi$ is roughly comparable as keeping the randomly initialized features fixed. Our rationale for fixing the features is as follows:
> As shown in [1], fixing randomly initialized features gives comparable results as with training features for ICM. From our experience with the commonly adopted way of training the SF with an autoencoder structure, we experienced that the interleave between training the two objectives (learning the features and rewards, learning the SF) is relatively sensitive to hyperparameter settings. Due to those reasons we wanted to investigate the possibility of minimizing the procedure and the computation budget required for learning the SF, just as [1] investigated the possibility of using a fixed feature extractor. As the experiments suggested, this setup can serve our method for the experiments considered, as although randomly initialized, the CNN structure is already a good prior for feature extractors at least in the visual input domain.
>
> $\bullet$ Comment:
> “It wasn't clear from the exposition which policy is used to train the SFs? The exploratory policy, the uniform random one or the behavior policy (the combination between the two trained policies given by the Q-nets).”
>
> Response:
> Samples from the replay buffer, which is filled with the experiences from the behavior policy, is used to train the SFs. We have clarified that in the new version.
>
> $\bullet$ Comment:
> “On the SID setup. Did you conduct any studies on M (the number of switches)? For instance, how does this compare with something like episode switching, which has been explored before?”
>
> Response:
> This is a very interesting point. We conducted a small experiment on MyWayHome with different choices for the number of switches per episode and have added the results to the appendix (Appendix B). The results show that our method is not very sensitive about the exact number of switches, however in this environment the episode switching performed worse.
> We suspect that since our scheduler schedules several times (e.g. 8 times) during an episode, and that we have 2 sub-policies to choose from, the behavior policy space is implicitly increased by exponential to 2^8 which can greatly increase the behavior diversity and helps exploration. We added more discussion of the scheduling choices in Appendix F.
>
> $\bullet$ Comment:
> There are a couple of observation/discussion claims that are not really substantiated (for instance, last paragraph in Sec. 3.2).
>
> Response:
> We agree with the reviewer and revised that.
>
> $\bullet$ Comment:
> "The link to the video (https://gofile.io/?c=HpEwTd .) doesn't work. Please update."
>
> Response:
> We thank the reviewer for the hint. At the current stage we are unable to edit the abstract in openreview. The correct url is https://gofile.io/?c=HpEwTd and we are sorry for the inconvenience.
>
>
> [1] Yuri Burda, Harri Edwards, Deepak Pathak, Amos Storkey, Trevor Darrell, and Alexei A Efros. Large-scale study of curiosity-driven learning.arXiv preprint arXiv:1808.04355, 2018a.

---

### Official Review · AnonReviewer4 · 2019-11-05
**Official Blind Review #4**

**Rating:** 3

**Review:**


## Summary

This paper proposes a novel intrinsic reward for exploration called SFC (successor feature control), to deal with sparse-reward and hard-exploration task. The main idea of SFC is to provide an agent with intrinsic reward defined to be the L2 distance between the successor features of two consecutive states (Equation 4). An underlying motivation of SFC exploration is that high SFC would encourage the agent to enter the "bottleneck states" and therefore helps to explore the entire state space.

Another line of contribution is SID (scheduled intrinsic drive), where a scheduler is used to determine which of the two separate policies (one for extrinsic reward and intrinsic reward) is chosen, with a fixed probability in their implementation, and executed for the next rollout of experience. It has an effect of longer-term exploration and prevents the agent from collapsing to a local-optimum behavior.

Empirically, the SFC+SID algorithm is evaluated on custom sparse-reward navigation-type environments such as VizDoom and DeepMind Lab (as well as a simple pixel-based continuous control), and outperforms other intrinsically motivated RL algorithms including RND and ICM.



## Overall Assessment

Overall, I like the idea of this paper and finds it very interesting and promising, but feel it would be on the borderline or slightly below the bar.

This paper studies a very interesting and novel approach of leveraging successor features for exploration. Successor features are a promising way of learning dynamics-related, task-agnostic representation for RL, which can provide a temporally extended exploration signal. The resulting method presents an improvement over existing intrinsic-reward exploration algorithms.

However, I think there are some weaknesses of the paper that would put the paper slightly below the acceptance threshold: empirically the environments are not diverse enough, and they have an implicit structure assumed and favorable for the proposed method --- there remains a question whether the method is general and not task-specific. I also think there are some misleading overclaims. Please see the detailed comment below.



## Detailed Comments

**[Problem Motivation and Significance]**
This paper address an important, long-standing problem in RL of efficient exploration under sparse-reward environments.

A minor comment: in the introduction, it is said that "terminal reward RL settings" are considered (the reward is given when the goal is achieved) --- which is an extreme case of sparse-reward RL problems --- but in the experiments non-terminal reward environments are studied, e.g. "AppleDistractions" where each of apples yields +0.05 reward. I think the overall claim could be a bit toned down to, for instance, dealing with sparse-reward environments.

**[Clarity]**
Overall the paper is clearly written and easy-to-follow. Descriptions of implementation details are well provided. However, there are some parts that can be better clarified and improved more. Please see more detailed comments below.

**[Justification of Method (SFC & SID)]**

The choice of successor feature for driving a novelty-like intrinsic reward signal seems well-motivated. This is because learning of successor features is task-agnostic and only related to multi-step dynamics (though SF is with respect to "a policy"), which gives a good representation that captures topological characteristics of the environment. The way intrinsic reward is derived, the squared distance of SFs of $S_t$ and $S_{t+1}$, basically encourages the agent to visit and go across the "bottleneck" state.

In the description of methods, it should be clearly noted that what is the underlying policy being learned for deriving successor features (i.e. $\pi$ in Eq.3 and 4) --- for example, is it a behavoral policy (which is a mixture of two policies) induced by scheduled drive? Another comment related to this about "SFC captures statistics over the full distribution of policies that have been followed, ..." (section 2), which sounds a bit overclaiming to me. Please note that a SF is with respect to a specific policy (e.g. behavioral policy), from the expectation in the definition; I think the use of past experience for minimizing the TD error is basically for estimating the expectation term through approximation, so I am not sure that this claim is well-justified.

It is not very clear to me why the SFC reward agrees with bottleneck states. I don't think the explanation given in Section 3.2 is logically enough. Also, isn't it only true under a random exploration policy? How would you defined the "bottleneck states" (e.g. Tomar et al. 2019) -- which can be helpful for making the main idea more understandable? Moreover, there was no enough explanation or reasoning about why SD (successor distance) is roughly the shortest path between the states.

In SID, a policy for extrinsic rewards and another policy for intrinsic one are learned. But in case the extrinsic reward was never received (in case of terminal-reward environments), the former policy would behave no different than random policies. Is this interpretation correct?

Also, given the presented form of SID, it sounds like a bit overclaiming to say it is a hierarchical RL agent, since the scheduler just picks one of the policies with equal probability (rather than being learned) --- especially one policy would become a random exploration policy --- and there is no notion of abstraction or goal/options.



**[Environment choice]**
I feel (1) that the environments being evaluated on are not diverse enough, and (2) that the environment in the experiment seems to exhibit specific properties that are favorable to the algorithm.

(Bottleneck State) One implicit assumption is that the structure of navigation is chosen such that following bottleneck states would lead to an optimal trajectory. I agree that even on the maze like FlytrapEscape the navigation/exploration problem is not easy in the absence of rich reward signals, but this is exactly a sort of environments on which SFC can perform better, especially compared to RND/ICM which are not attracted by bottleneck states (Appendix A). It is good though, and could be beneficial in many cases with the presence of bottleneck states, but seems general applicability is a little bit short (not as much as claimed).

(Distinctive appearance) Another assumption is about a choice of appearance. One important thing to note about SF learning is that a feature for state or transition (cumulant) is kept fixed after random initialization, rather than being learned as in (Machado et al. 2018; Kulkarni et al. 2016; Barreto et al. 2017). This is because this method does not need to do regression of reward function. Then, the state-feature $\phi(s)$ should be discriminative enough so that it can capture some topological and global characteristic of the state space. In general, this is not an easy problem (for first-person view POMDPs), but seems on the environments (FlytrapEscape, AppleDistractions) it was possible because each room/sector has uniquely identifiable wall color and texture. I feel this is somewhat strong assumption made to make SF work. Thus, "We believe this is the first time that SF are shown to behave in a first-person view environment as one would expect from its definition" would sound a bit overclaiming. Would this method work on more general environments that do not have this property --- specifically, what will happen if rooms are not distinguishable from color and texture (and the walls were looking similar)?

Control from pixels (DM Cartpole) is an example of environment that does not have these assumptions, but one downside is that action space was simplified and discretized. Indeed, the improvement shown on Cartpole over ICM/RND is not substantial enough. To demonstrate that SFC+SID is "generally useful" as claimed in the paper, presenting benchmark results on standard discrete-action Atari environments, or more diverse RL environments would have greatly strengthened the paper to be more convincing.



**[Analysis of successor distance]**
Figure 11 (visualization of successor distance) is a great analysis, and I liked it. It clearly shows a smooth topology of the environment thanks to the temporally-extended representation that SF captures. I found that the difference heatmap is a little bit difficult parse. Also, under which policy the SF was computed (I guess this is a behavior policy derived by SID; it should be clearly mentioned somewhere in the paper)?

**[More minor comments about experiments]**
- Was the same K-step objective (e.g. K=5) used for all of SFC, ICM and RND? If so, what would the result look like when K=1?
- The ablation study (appendix 1) is interesting and very important. The "Ours" algorithm in the main text is actually a combination of SFC and SID, so the comparison shown in this ablation study could be a main result.



## Feedback for Improvement

More related work:

* Learning decomposed value functions for extrinsic and intrinsic rewards have been discussed in (Burda et al, 2018b), though in their work a single policy is being learned.
* [Comparison with Machado et al. 2018] It is discussed that (Machado et al. 2018: count-based exploration with SR) is very similar because of the use of SR/SF. The ways of how to derive intrinsic reward signal are indeed different, but it would be great to have a detailed discussion about how they are different or similar.



Minor comments:

* Citation needed on section 3.1 --- (Kulkarni et al. 2016 or Barreto et al. 2017)
* Please consider putting the environment name in the title of each learning curve.
* Typo: Therfore (right before section 3.2)
* Typo: temporarily -> temporally (introduction bullet point 2)



**Experience Assessment:**

I have published one or two papers in this area.

**Review Assessment: Checking Correctness Of Derivations And Theory:**

I carefully checked the derivations and theory.

**Review Assessment: Checking Correctness Of Experiments:**

I carefully checked the experiments.

**Review Assessment: Thoroughness In Paper Reading:**

I read the paper thoroughly.

---

> ### Author Response · Authors · 2019-11-12
> **Response to Reviewer #4: Justification of Method (SFC)**
>
>
> $\bullet$ Comment:
> “In the description of methods, it should be clearly noted that what is the underlying policy being learned for deriving successor features (i.e. in Eq.3 and 4) --- for example, is it a behavioral policy (which is a mixture of two policies) induced by scheduled drive?”
>
> Response:
> The replay buffer stores the experience generated from the behavior policy and the SF-net are trained with samples from that buffer. We clarified that in the new version.
>
> $\bullet$ Comment:
> “It is not very clear to me why the SFC reward agrees with bottleneck states. I don't think the explanation given in Section 3.2 is logically enough. Also, isn't it only true under a random exploration policy? How would you define the"bottleneck states?
>
> Response:
> With bottleneck states we mean states where a minor change in its action selection would result in vastly different future trajectories. This concept of bottlenecks motivated directly the definition of our intrinsic reward bonus.
> To give an intuitive explanation: The SF represent a state by a function of its successor states. So for states that lie within a well connected region in the state space (e.g. a room), their future most likely look quite similar (there is a high chance that their consecutive states lie in the same room). This means that the SFC reward for these states is low. On the other hand the future trajectory of state at doors/exits are sensitive to the choice of the next action, because it determines whether the agent ends up in a different room or not. Therefore these states corresponds to a high SFC reward.
>
> Fig. 1 b) of the paper shows that this intuition is correct at least in an established bottleneck toy problem. It is true that the reward here depends on the policy the SF were learned from (random agent here). But  as long as the SF are learned under a sufficiently stochastic policy is expected to have similar properties. This stochasticity is well satisfied under the SID framework, as whenever the extrinsic policy is scheduled, the agent acts pseudo randomly before receiving external reward.
>
> We also note that the SFC reward was inspired by the definition of bottlenecks, but is not limited to environments with structural bottlenecks. By definition the SF captures the transition dynamics and the dynamics induced by the policy. This means that in addition to the environmental bottlenecks (induced by the transition dynamics of the environment), SFC is also capable to capture “perceived bottlenecks” (induced by the policy: ). This means, also as shown by our cartpole experiment, that SFC is still able to bring performance gain in environments without apparent environmental bottlenecks, since detecting “perceived bottlenecks” (where the agent also receives high SFC reward) induced by the behavior policy can help to avoid well explored regions of the state space.
>
> $\bullet$ Comment:
> “In SID, a policy for extrinsic rewards and another policy for intrinsic one are learned. But in case the extrinsic reward was never received (in case of terminal-reward environments), the former policy would behave no different than random policies. Is this interpretation correct? “
>
> Response:
> Roughly that is correct. However, the action is still the output of a network that is trained with a TD loss which is not exactly zero because of the bootstrapping term although the rewards are zero. Depending on the initialization and the visited states so far the distribution of actions coming from the network might be biased.
>
> $\bullet$ Comment:
> “Figure 11 ...under which policy the SF was computed (I guess this is a behavior policy derived by SID; it should be clearly mentioned somewhere in the paper)?”
>
> Response:
> The visualized SF are the result from training an agent with SID and SFC (i.e. the SF are trained with all experience collected while scheduling between the extrinsic policy and the SFC policy). We have clarified that in the new version.
>
> $\bullet$ Comment:
>  “Another comment related to this about "SFC captures statistics over the full distribution of policies that have been followed, ..." (section 2), which sounds a bit overclaiming to me. Please note that a SF is with respect to a specific policy (e.g. behavioral policy), from the expectation in the definition; I think the use of past experience for minimizing the TD error is basically for estimating the expectation term through approximation, so I am not sure that this claim is well-justified.“
>
> Response:
> We agree that the formulation can be misleading and have revised it.

---

> ### Author Response · Authors · 2019-11-12
> **Response to Reviewer #4: Environment Choice**
>
>
> $\bullet$ Comment:  Environments with distinctive appearance
>
> Response:
> This is a very insightful point. Actually, in MyWayHome which is used as a common benchmark in several intrinsic motivation papers, each room has distinctive textures. And since the formulations of ICM, RND or SFC depend very much on the features of the textures it would cause those methods trouble if the textures are the same in all rooms. We suspect that same texture would cause trouble for most navigation algorithms that are only vision-based, and it is not a limitation only to SFC. We hypothesis that having recurrent structures in the network or having an auxiliary task to regress the depth could help but that falls out of the focus of this paper.
> And we note in AppleDistractions the textures for each of the corridors are randomly generated for each of the three environments, so the texture is not necessarily distinct for each corridor.
>
> $\bullet$ Comment:
> “the environments being evaluated on are not diverse enough”
>
> Response:
> To evaluate our method on more diverse environments we conducted preliminary experiments on Montezumas Revenge. We are aware that many works proposed to solve this game, but we also speculate that this environment poses a special kind of exploration challenge, where one uncareful exploration move would make the agent die.
> We can not provide conclusive results on this environment given the limited time and computing resources. However, our preliminary results show that our method is general enough to also help in this environment, while we do not claim that our method achieves state-of-the-art results on Montezuma. A plot of the results can be found under https://gofile.io/?c=0XM4LG .
> Because of the time constraints we could only test our method using SFC and SID against a pure extrinsic reward agent. The base RL algorithm was again Ape-X with 8 actors with a replay memory of 500k, run for 400k gradient updates which corresponds to roughly 200 million frames.
>
> $\bullet$ Bottleneck State Structure
>
> Response:
> We note that MyWayHome is a common benchmark. Furthermore, Cartpole does not contain apparent bottlenecks and we also provide preliminary experiments on Montezuma's Revenge. All in all, we demonstrate that our method works well on a wide variety of environments with different properties.
>
> $\bullet$ Discretization of continuous control in DM Cartpole
>
> Response:
> In many continuous control tasks discretization is avoided because the search space blows up exponentially with the number of action dimensions (curse of dimensionality). In this specific task it is not the case (action dimension = 1) and it can be solved even with a discretized action space. We note that we used one of the simplest discretization procedures (linear spacing) and this same action space is not just used for our method but for all baselines, too.
> We test our method in the cartpole environment, because it is a sparse reward task but differs much from our previous tasks in the sense that it has no obvious bottleneck states and that observations are pretty similar across the whole state space, and the fact that it is third-person view instead of first-person. The only reason we discretized the state space was due to the use of APE-X DQN as our backbone, we did not want to prove properties of our method specific to continuous control as that is not the focus of our paper.

---

> ### Author Response · Authors · 2019-11-12
> **Response to Reviewer #4: Justification of Method**
>
> We thank the reviewer for the remarks and detailed suggestions for improvement.
>
> $\bullet$ Comment:
> The ablation study (appendix 1) is interesting and very important. The "Ours" algorithm in the main text is actually a combination of SFC and SID, so the comparison shown in this ablation study could be a main result.
>
> Response:
> We thank the reviewer for the helpful suggestions and have revised section 3.1 accordingly in the new version.
>
> $\bullet$ Comment:
> “Also, given the presented form of SID, it sounds like a bit overclaiming to say it is a hierarchical RL agent, since the scheduler just picks one of the policies with equal probability (rather than being learned) --- especially one policy would become a random exploration policy --- and there is no notion of abstraction or goal/options.”
>
> Response:
> We designed and investigated different kinds of high-level schedulers (e.g. a macro-Q scheduler), other than the random one, which we listed in Appendix F. We note that hierarchical refers to methods that involves a chained level of decision making, as in this paper we have a high-level scheduler and low-level sub-policies.
> We agree with the reviewer that our hierarchical setup is quite different from the previous options framework, where the optimal policy is constructed by a set of options. But there are different pathways to achieve hierarchical. Other than the options framework, “Learning by Playing” (Riedmiller 2018) proposes another line of hierarchical RL setup which inspired our work. While their method schedules between the extrinsic task and a set of pre-defined auxiliary tasks such as maximizing touch sensor readings or translation velocity, in this paper we adapted it to find an alternative solution to utilizing intrinsic motivation other than the commonly adopted reward bonus setup.
> We note that our scheduler schedules several times (e.g. 8 times) during an episode. Since we have 2 sub-policies to choose from, the behavior policy space is implicitly increased exponentially to 2^8 which can greatly increase the behavior diversity and helps exploration. We added more discussion of the scheduling choices in Appendix F.
> For the scheduler, as we discussed in Appendix F, we chose the random one as empirically none of the other schedulers we tried consistently outperform the random one. By Occam’s razor we present the random one as in our final method. But instead of viewing it as a disadvantage, we kindly invite the reviewer to see it as an advantage of our method, that the simplest scheduling design is able to bring perform gain.
>
> $\bullet$ Comment:
> “Was the same K-step objective (e.g. K=5) used for all of SFC, ICM and RND? If so, what would the result look like when K=1?”
>
> Response:
> We used K=5 for all agent configurations. As reported in the Ape-X paper the 5-step in general is much more efficient than other settings. We did preliminary experiments with 1,3,5 step sizes in “my_way_home” and concluded the same. Thus to save the overall computation time we choose the most time efficient setup, which is also the default setup suggested by Ape-X.
>
> $\bullet$ Comment:
> “One important thing to note about SF learning is that a feature for state or transition is kept fixed after random initialization, rather than being learned“
>
> Response:
> In our preliminary experiments we found that the performance of training the feature embedding $\theta_\phi$ is roughly comparable as keeping the randomly initialized features fixed. Our rationale for fixing the features is as follows:
> As shown in [1], fixing randomly initialized features gives comparable results as with training features for ICM. From our limited experience with the commonly adopted way of training the SF with an autoencoder structure, we experienced that the interleave between training the two objectives (learning the features and rewards, learning the SF) is relatively sensitive to hyperparameter settings. Due to those reasons we wanted to investigate the possibility of minimizing the procedure and the computation budget required for learning the SF, just as [1] investigated the possibility of using a fixed feature extractor. As the experiments suggested, this setup can serve our method for the experiments considered, as although randomly initialized, the CNN structure is already a good prior for feature extractors at least in the visual input domain.
> For environments with the same textures everywhere, we suspect that training the features for SF would also not necessarily help, as with the same input the feature embedding would give one embedding no matter trained or fixed.
>
>
> [1] Large-scale study of curiosity-driven learning, Burda et al., 2018a

---

### Author Response · Authors · 2019-11-12
**Overview of revisions in the updated draft**

We thank all the reviewers for their time and effort in reviewing our paper. We appreciate the positive feedback on our proposed method, we also appreciate the constructive feedback on how to further improve our paper with the detailed reviews. We uploaded a revised version incorporating the suggestions of reviewers, we list the main revisions below.

$\bullet$ As suggested by several reviewers, we incorporated contents from Appendix A, where ablation studies are presented, to the main paper.
$\bullet$ We conducted an additional experiment to examine how the number of switches per episode in SID affects the performance and report the results in the Appendix.
$\bullet$ In the previous version, we missed a reference to the Appendix F where we presented several high-level schedulers that we had investigated. We added the missing reference. Also we included more detailed discussion about the scheduler.
$\bullet$ We fixed typos and clarified potentially misleading statements.

---

### Decision · Program_Chairs · 2019-12-19

**Decision:**

Reject

**Comment:**

The paper presents a method for intrinsically motivated exploration using successor features by interleaving the exploration task with intrinsic rewards and extrinsic task original external rewards. In addition, the paper proposes "successor feature control" (distance between consecutive successor features) as an intrinsic reward. The proposed method is interesting and it can potentially address the limitation of existing exploration methods based on intrinsic motivation. In experimental results, the method is evaluated on navigation tasks using Vizdoom and DeepMind Lab, as well as continuous control tasks of Cartpole in the DeepMind control suite, with promising results.

On the negative side, there are some domain-specific properties (e.g., moderate map size with relatively simple structures, different rooms having visually distinct patterns, bottleneck states generally leading to better rewards, etc.) that make the proposed method work well. In addition, off-policy learning of the successor features could be a potential technical issue. Finally, the proposed method is not evaluated against stronger baselines on harder exploration tasks (such as Atari Montezuma's revenge, etc.), thus the addition of such results would make the paper more convincing. In the current form, the paper seems to need more work to be acceptable for ICLR.